# Dynamics of avalanche-generated impulse waves: three-dimensional hydrodynamic simulations and sensitivity analysis

Rachel E. Chisolm[1], Daene C. McKinney[1]

[1]Department of Civil Architectural and Environmental Engineering, University of Texas at Austin, Austin, TX, USA

*Correspondence to*: Rachel E. Chisolm (rachel.chisolm@gmail.com)

Keywords: Glacier lake outburst flood, Glacial lake, Lake hydrodynamics, Lake Palcacocha, Peru, Avalanche-generated waves

**Abstract.** This paper studies the lake dynamics for avalanche-triggered glacial lake outburst floods (GLOFs) in the Cordillera Blanca mountain range in Ancash, Peru. As new glacial lakes emerge and existing lakes continue to grow, they pose an increasing threat of GLOFs that can be catastrophic to the communities living downstream. In this work, the dynamics of
displacement waves produced from avalanches are studied through three-dimensional hydrodynamic simulations of Lake Palcacocha, Peru, with an emphasis on the sensitivity of the lake model to input parameters and boundary conditions. This type of avalanche-generated wave is an important link in the GLOF process chain because there is a high potential for overtopping and erosion of the lake-damming moraine. The lake model was evaluated for sensitivity to turbulence model and grid resolution, and the uncertainty due to these model parameters is significantly less than that due to avalanche boundary
condition characteristics. Wave generation from avalanche impact was simulated using two different boundary condition methods. Representation of an avalanche as water flowing into the lake generally resulted in higher peak flows and overtopping volumes than simulating the avalanche impact as mass-momentum inflow at the lake boundary. Three different scenarios of avalanche size were simulated for the current lake conditions, and all resulted in significant overtopping of the lake-damming moraine. Although the lake model introduces significant uncertainty, the avalanche portion of the GLOF process chain is likely
to be the greatest source of uncertainty. To aid in evaluation of hazard mitigation alternatives, two scenarios of lake lowering were investigated. While large avalanches produced significant overtopping waves for all lake-lowering scenarios, simulations suggest that it may be possible to contain waves generated from smaller avalanches if the surface of the lake is lowered.

## 1 Introduction

Glacier retreat worldwide has resulted in the emergence and growth of glacial lakes that have replaced ice in the tongue area
of many glaciers, and a large number of these lakes pose a hazard or risk of glacial lake outburst floods (GLOFs). GLOFs are common in many parts of the world, and they can be catastrophic to downstream communities and infrastructure. Emmer et al. (2016a) have compiled a worldwide database of GLOF events, including approximately 20 events in the Peruvian Andes.

Wang et al. (2015b) found that glacial lakes in the central Himalaya have expanded significantly (122.1%) from 1976 to 2010, and and Schwanghart et al. (2016) showed that more than 68% of Himalayan hydropower projects are located on potential GLOF tracks. Allen et al. (2016) performed a first-order assessment of GLOF risk across the Himalayan state of Himachal Pradesh (HP), Northern India, including locations where future lakes might form. They identified areas with potentially high

GLOF risk and determined that GLOF hazard is likely to increase in the future with continued deglaciation. Linsbauer et al. (2016) calculated glacier overdeepenings and predicted the emergence of future lakes in the Himalaya and Karakoram in relation to GLOF risk and found approximately 5000 overdeepening locations that could form significant glacial lakes. Cook et al. (2016) studied glaciers in the Bolivian Andes where glaciers have receded about 40% between 1986 and 2014, resulting in an increasing number of proglacial lakes. They identified 25 lakes that pose a potential GLOF threat to downstream

communities and infrastructure. The Cordillera Blanca mountain range in Peru has approximately 1900 lakes, and 830 of them have a surface area greater than 5000 $m^2$ (UGRH, 2014). Of the lakes in the Cordillera Blanca, over 200 are considered new lakes that have formed recently due to glacier retreat (UGRH, 2014). A recent inventory of glacial lakes in the Cordillera Blanca and their susceptibility to GLOFs classifies Lake Palcacocha in the highest level of susceptibility to outburst floods due to mass movements into the lake (Emmer et al., 2016c). Several GLOFs have occurred in the Cordillera Blanca in recent

history, and climate change and accelerated glacial retreat have been increasing the GLOF hazard since the end of the Little Ice Age in the late 1800's (Carey, 2010).

GLOFs can be highly destructive because the peak discharges tend to be several orders of magnitude larger than typical outflows from glacial lakes (Benn and Evans, 2010). Moraine-dammed lakes are particularly susceptible to outburst flooding due to the potential for moraine failure that could cause higher GLOF discharges than would occur with just overtopping

(Emmer and Cochachin, 2013); however, both moraine-dammed and bedrock-dammed lakes can produce potentially catastrophic GLOFs due to overtopping if there is insufficient freeboard. According to studies that have established basic methods for evaluating potential glacial lake hazards (e.g., Haeberli et al., 1989; Huggel et al., 2004; Wang et al., 2011; Emmer and Vilímek, 2014; Rounce et al., 2016), the primary characteristics that signify a potentially hazardous glacial lake are the presence of overhanging ice and the likelihood of failure of the lake-damming moraine; secondary characteristics that indicate

potentially dangerous lakes include the potential for landslides, rock-slides or rock-ice avalanches. However, understanding of the physical processes that can trigger a GLOF event is still limited. The most common GLOF triggers are landslides, avalanches, or ice calving into a lake (Costa and Schuster, 1988; Richardson and Reynolds, 2000; Bajracharya et al., 2007; Awal et al., 2010; Emmer and Cochachin, 2013; Emmer and Vilímek, 2013). These mass movement events can cause large waves that propagate across glacial lakes and may overtop or breach moraine dams.

Several studies have looked at GLOF events after they have happened and attempted to reconstruct the GLOF characteristics. Worni et al. (2014) and Westoby et al. (2014a) review various methods for modeling a typical GLOF process chain. Some researchers have simulated GLOFs with models of the individual processes in the chain (e.g., Klimes et al., 2014; Schneider et al., 2014; Westoby et al., 2014b; Worni et al., 2014; Wang et al., 2015a; Somos-Valenzuela et al., 2016); however, the lake dynamics remain one of the most difficult processes to simulate correctly due to the need for a non-hydrostatic model to

correctly represent the wave dynamics (Heinrich, 1992; Zweifel et al., 2007; Worni et al., 2014) and the lack of field data for model calibration or validation. Most previous studies have used two-dimensional shallow water equation models (2D SWE models) or empirical models of wave generation and propagation in the lakes that do not effectively represent the physical processes, and our understanding of the uncertainties arising from avalanche-generated wave simulations in GLOF process chain modeling is still very limited. Although they have seldom been studied, evaluating the sensitivities of displacement wave models helps us understand the uncertainties that may arise from wave simulations within GLOF process chain modeling. One difficulty is the lack of data about real events, so the potential hazard and impacts of a GLOF must be estimated from an analysis of the physical conditions and modeling the basic physical processes without the availability of calibration data (Somos-Valenzuela et al., 2016). This paper presents three-dimensional non-hydrostatic simulations of avalanche-generated waves and improves upon previous two-dimensional SWE simulations of avalanche-generated waves in GLOF process chain modeling that must be calibrated with data from past GLOF events (e.g., Schneider et al., 2014). Many glacial lakes that are currently dangerous have not previously outburst, so the use of data from prior GLOF events is not an option at many study sites. Three-dimensional non-hydrostatic models provide better representation of the physical processes within the model than 2D shallow water models, thus reducing the amount of calibration needed; but, 3D non-hydrostatic models have rarely been applied to avalanche-generated waves and GLOF process chain modeling.

The objective of this paper is to gain a better understanding of the behavior of avalanche-generated waves and the factors that influence overtopping discharges. This paper evaluates the relative uncertainties generated in the lake model portion of the GLOF process chain with the goal of improving GLOF hazard assessments. Particular emphasis is placed on analysis of the sensitivity of the lake model to various input parameters and methods for simulating boundary conditions with the goal of shedding light on potential sources of uncertainty in the lake model. An improved understanding of the dynamics of avalanche-generated waves can help advance predictive modeling of potential GLOF events by relying less on model calibration when data from past events are not available, thus enabling better evaluation of possible hazard mitigation strategies at potentially dangerous lakes. It is a significant challenge to predict the impacts of an event that has not yet happened, and predictive simulations inherently carry considerable uncertainty about many event parameters. Nevertheless, this challenge is one that must be undertaken for progress to be made in glacial hazard assessment and analysis of hazard mitigation strategies.

**1.1 Lake Palcacocha**

GLOFs have been a problem in the Cordillera Blanca for many years (Lliboutry, 1977; Reynolds, 2003; Carey, 2010). The most disastrous GLOF event in the Cordillera Blanca in recent history occurred in 1941 when Lake Palcacocha burst, destroying much of the city of Huaraz and killing approximately 1800 people (Carey, 2010; Wegner, 2014). This event received much notice from national and international media and put the issue of GLOFs at the forefront of national attention in Peru. Huaraz is the most populous city in the Cordillera Blanca region with over 100,000 residents (INEI, 2007 census), and it is once again exposed to a potential GLOF from Lake Palcacocha (Somos-Valenzuela et al., 2016). After the 1941 Huaraz flood, the Peruvian government instituted initiatives to reduce the GLOF risk in the Cordillera Blanca through monitoring of glaciers

and glacial lakes and implementing lake safety systems (Carey, 2010). These safety systems typically consist of tunnels to control lake levels, reinforced dams or a combination of the two (Portocarrero, 2014). Scientists and engineers in Peru have several decades of experience managing glacial lakes in the Cordillera Blanca and mitigating GLOF risk (Carey, 2010; Portocarrero, 2014), but current lake management practices are based on studies performed decades ago that have not been

updated to account for changes that have occurred since then, primarily increased size and water storage in glacial lakes due to changing climate. The lake safety system implemented at Lake Palcacocha in the 1970's was designed for the size of the lake at the time and did not account for future lake growth. If the present knowledge of climate change existed at that time, perhaps this could have been foreseen; this was not the case, and now the lake is approximately 17 times larger than it was in 1974 (Rivas et al., 2015), rendering the existing lake safety system inadequate for the current lake dimensions (Portocarrero,

2014). The potential threat that Lake Palcacocha currently poses to the residents of Huaraz has been known for many years. Peruvian government institutions have produced several official reports about the situation (INDECI, 2011; ANA, 2013; Valderrama et al., 2013; Espinoza, 2013; INDECI, 2015), and a state of emergency was declared in 2010 (Diario la Republica, 2010; INDECI, 2011). In this paper, Lake Palcacocha is used as a case study to investigate the impact of an avalanche event on the lake dynamics and the ensuing discharge hydrograph from the lake and to study the sensitivity of the overtopping

discharge to various input parameters. This paper focuses exclusively on the lake dynamics for avalanche-generated waves and does not consider other parts of the GLOF process chain. The model results for the entire GLOF chain of events are presented in Somos-Valenzuela et al. (2016).

Lake Palcacocha (4562 m) is situated in the Quillcay watershed above the city of Huaraz (Fig. 1). Above the lake are the Palcaraju and Pucaranra glaciers. The steep overhanging ice of the glacier termini in contact with the lake makes it extremely

prone to avalanche-generated waves. Additionally, the large volume of water contained in the lake provides a serious threat to downstream areas. The lake is surrounded on three sides by glacial moraines, and the lateral moraines are very tall with slopes up to 80º (Klimes et al., 2016). The southern lateral moraine is prone to landslides into the lake, and a slide from this moraine in 2003 caused minor damage from a wave that overtopped a portion of the lake-damming moraine (Vilimek et al., 2005). The original lake-damming terminal moraine was breached during the 1941 GLOF, and the lake is currently dammed by a smaller

basal moraine that lies about 300 m upstream of the 1941 breach. This smaller moraine that currently holds back the lake is approximately 66 m deep, 985 m wide and has a width-to-height ratio of 14.9; this morphology indicates that the lake-damming moraine is very stable (Rivas et al., 2015). A constant lake level of 4562 m (8 m of freeboard) is maintained by a structure in the smaller terminal moraine that consists of an open cut in a portion of the moraine filled with an artificial dam that was constructed in 1974 (Reynolds, 2003; Portocarrero, 2014). Additionally, two sections of the terminal moraine have been

reinforced with concrete to protect them from erosion. Based on a 2009 bathymetric survey, the volume of the lake was approximately 17 million m$^3$ at that time (UGRH, 2009). The lake has since retreated approximately 200 m more (Rivas et al., 2015), and siphons are currently being used to temporarily maintain the lake an additional 3-5 m lower; however, during the rainy season, the siphon system is often not able to keep up with the rainfall draining into the lake. A bathymetric survey undertaken in February, 2016, measured a volume of approximately 17.4 x 10$^6$ m$^3$ with a water surface elevation of 4562.88

m (UGRH, 2016). Lake Palcacocha has a deep area adjacent to the glacier with a maximum depth of 72 m and a shallow portion with depths mostly under 10 m extending several hundred meters back from the terminal moraine (Fig. 2).

The potential hazard due to an outburst flood from Lake Palcacocha has been studied by several researchers. Vilimek et al. (2005) discussed the influence of glacial retreat on hazards at Palcacocha and studied the moraine composition and the potential for landslides from the lateral moraines; they also found seepage at the moraine dam. Emmer and Vilimek (2013) used a generalized methodology for GLOF hazard assessment at Lake Palcacocha and 5 other lakes in the Cordillera Blanca, concluding that Lake Palcacocha had the highest hazard level. Emmer and Vilimek (2014) examined mechanisms of the 1941 and 2003 GLOFs at Lake Palcacocha and compared them to other historic GLOFs in the Cordillera Blanca. Emmer et al. (2016b) evaluated the effectiveness of lake safety systems in the Cordillera Blanca and found that the system at Lake Palcacocha resulted in a minimal decrease in GLOF susceptibility, and Emmer et al. (2016c) classified Lake Palcacocha as highly susceptible to GLOFs triggered by mass movements. Klimes et al. (2016) evaluated the lateral moraines surrounding Lake Palcacocha and determined that there is a high potential for landslide-triggered waves in the lake. Rivas et al. (2015) modeled a full moraine collapse using empirical equations and DAMBRK hydraulic simulations, and Somos-Valenzuela et al. (2016) gave the results of simulations of a potential GLOF chain of events and mapped potential hazard levels for the city of Huaraz. This paper focuses on the avalanche boundary conditions, turbulence modeling and grid size and their relative contributions to uncertainty in the lake model used by Somos-Valenzuela et al. (2016) to model the entire GLOF process chain and downstream impacts.

## 1.2 Impulse Waves Generated from Avalanches and Landslides

The dynamics of avalanche or landslide-generated waves are very complex (Fritz et al., 2004; Worni et al., 2014). In addition, it is very difficult to obtain field measurements of these waves to better understand their dynamics, and most of the data from actual events are estimates based on residual evidence in the field (e.g., run-up on side slopes or moraine erosion). The physical principles governing the mechanics of wave generation and propagation are presented in Dean and Dalrymple (1991). A number of studies have developed empirical models from laboratory simulations and/or field data of avalanche and landslide generated waves (e.g., Kamphuis and Bowering, 1970; Slingerland and Voight, 1979 and 1982; Fritz et al., 2004; Heller and Hager, 2010), but many of the laboratory models use simplified geometries (Heller et al., 2016). Numerical simulations of slide-generated waves have been primarily focused on two-dimensional simulations and simple arrangements (e.g., Rzadkierwicz et al., 1997; Zweifel et al., 2007; Biscarini, 2010; Cremonesi et al., 2011; Ataie-Ashtiani et al., 2011; Ghozlani et al., 2013); but, the two-dimensional shallow water equations (SWE) may not be appropriate for slide-generated waves because of the role that vertical accelerations play in the wave dynamics (Heinrich, 1992; Zweifel et al., 2007). Recent developments in numerical simulations of landslide-generated waves include simulation of multi-phase flows, including a three-dimensional Navier-Stokes Volume of Fluid model (Abadie et al., 2010), a two-phase debris flow model (Kafle et al., 2016), and the application of Smoothed Particle Hydrodynamics (SPH) models (Heller et al., 2016; Wang et al., 2016). However, these studies still focus on simple cases and geometries rather than real-world scenarios. R.avaflow, a two-phase

model that was developed to simulate debris flows into fluid bodies (Pudasaini, 2012), has been applied to simulate debris flows such as those that would be present in a GLOF (Mergili et al., 2017). Few researchers have looked at the issue of wave run-up (e.g., Synolakis, 1987 and 1991; Muller, 1995; Liu et al., 2005; Capel, 2015; Romano et al., 2015; Etemad-Shahidi et al., 2016), and most use empirical formulas or simplified approaches for wave run-up calculations, making assumptions about the lake geometry that may not be realistic (e.g., uniform water depth and a regularly sloped dam).

Although models of real events are limited by the lack of validation data, there is clearly a need to move away from simplified cases such as sliding blocks or wedges and progress towards modeling cases that more closely resemble geometries and circumstances in the field. Use of three-dimensional numerical modeling can improve simulations of avalanche-generated waves by avoiding some of the weaknesses of two-dimensional shallow water models. Some of the problems of modeling avalanche-generated impulse waves include: uncertainty in the make-up of the avalanche material (e.g., ratio of snow, ice and rock; density; viscosity) and representation of the mixing and momentum transfer when the avalanche material enters the lake.

## 2 Methods

A three-dimensional hydrodynamic model, FLOW 3D (Flow Science, 2012), was used to simulate waves generated from avalanches entering a glacial lake and investigate the dynamics of the wave generation, propagation, run-up and moraine overtopping. Because three-dimensional models have rarely been applied to avalanche-generated waves, there is very little information on appropriate input parameters and boundary conditions. Therefore, several input parameters have been varied in this study in order to analyze the model sensitivities and gain a better understanding of the impact of user-specified inputs on model results. The sensitivity to the turbulence model and grid size used in the simulations were investigated to determine how much these aspects of the model might contribute to the overall uncertainty. Another challenge of simulating avalanche-generated waves is appropriate representation of the avalanche entering the lake. Because there is very little knowledge about how to appropriately simulate the avalanche flow and impact with the lake, two different boundary condition methods were used in this work to help facilitate analysis of the sensitivity of the overtopping discharge to the avalanche boundary conditions. Wave generation and propagation were studied to gain insights about how this type of wave behaves and what type of model is needed (2D vs 3D and hydrostatic vs. non-hydrostatic) to accurately reproduce avalanche-generated waves of the magnitude typically seen in GLOFs.

A three-dimensional, non-hydrostatic model was chosen to give as realistic a simulation environment as possible. Although two-dimensional SWE models have been applied to simulations of avalanche-generated impulse waves (e.g., Heinrich, 1992; Zweifel et al. 2007), the size and characteristics of the waves indicate that a three-dimensional model may be more appropriate because of highly variable water depths, wave heights and vertical accelerations. Additional motivation for employing this model is the variable lakebed geometries of many glacial lakes that tend to have sharp discontinuities near their terminal moraines that could significantly affect wave propagation and run-up (e.g., Lake Palcacocha, as seen in Fig. 2). The lakebed

topography in the FLOW 3D model was taken from a 2009 bathymetric survey (UGRH, 2009), and the topographic model that was used is a 5 x 5 m resolution DEM from airborne LIDAR and stereo images (Horizons, 2013).

Three avalanche scenarios that represent a range of likely avalanche sizes were simulated in addition to two lake-lowering scenarios to evaluate hazard mitigation alternatives. The discharge hydrographs resulting from the overtopping waves were

the inputs for a debris flow model used to determine the potential impact for the city of Huaraz (Somos-Valenzuela et al., 2016).

## 2.1 Sensitivity Analysis: Turbulence Model and Grid Size

### 2.1.1 Sensitivity to Turbulence Model

The FLOW 3D simulations used a three-dimensional, non-hydrostatic numerical scheme and a re-normalization group (RNG)

turbulence model with a dynamically computed mixing length; although, several other turbulence models were also tested. Very little information exists regarding the effect of turbulence models on the outcome of simulations of avalanche-generated waves. Therefore, the primary objective of assessing the sensitivity to the turbulence model is to determine how much the choice of turbulence model might affect overtopping discharges and how much attention should be given to this parameter in the modeling process.

Turbulence models are mathematical representations of the dissipation of energy from turbulence within the hydrodynamic model that cannot be represented in the model's discretization of the Navier-Stokes equations. There are a number of approaches to modeling turbulence that range in complexity. The simplest approach is an eddy viscosity model, a type of Reynolds Averaged Navier-Stokes (RANS) model that uses a single parameter to represent all of the dissipation of energy that occurs at the sub-grid scale. Usually this parameter is tied to a length scale that describes the flow such as depth or wave height.

More complex RANS models can use multiple parameters to describe the turbulence (e.g., two-equation models) or vary the length scale within a simulation based on the local flow conditions (i.e. models that use dynamically-computed mixing lengths). The ability to have a variable length scale that is calculated by the model is useful when an appropriate length scale is unknown, as is the case with the type of avalanche-generated wave studied in this paper due to the rapidly changing characteristics of the flow. Large Eddy Simulation (LES) is a different approach to modeling turbulence from the RANS models. LES simulates the

largest scales of turbulence within the hydrodynamic model and uses a filter to remove the smaller scales, which are accounted for within the turbulence model. The filter size is linked to the model grid size, and additional numerical errors can be introduced due to the filter width. LES models may be viewed as a middle ground between RANS models and direct numerical simulations (DNS) that solve the Navier-Stokes equations directly for all scales of turbulent flow (Ferziger and Peric, 2002). In this work, the RNG-dynamic mixing length model was chosen as the baseline turbulence model because an appropriate

mixing length was unknown due to the highly variable nature of the flow, both spatially and temporally. The sensitivity of the simulations to the turbulence model was tested by running repeat simulations for seven different turbulence models in FLOW 3D, including: (1) RNG-dynamic mixing length (baseline model), (2) RNG-constant mixing length, (3) k-epsilon, (4) Prandtl

mixing length, (5) one-equation-constant mixing length, (6) large eddy simulation (LES), and (7) laminar flow. Simulations of models (2) – (7) were compared to the baseline model for a large avalanche ($3x10^6$ m$^3$) at the current lake level using the percent difference in maximum wave height, peak overtopping flow rate, and total overtopping volume. Additionally, the root-mean-square deviation (RMSD) between the results of the baseline and the other models was calculated at each time step for
the outflow hydrographs and the flow depth at each point within the lake.

Turbulence models (1) – (5) are Reynolds-averaged Navier-Stokes (RANS) eddy viscosity models (Pope, 2000). Model (2) is a variant of model (1) except that it uses a constant mixing length (Yakhot and Orszag, 1986). Model (3) is a two-equation model that uses several standard constants. Models (4) and (5) are the simplest eddy viscosity models used. In FLOW 3D, the constant mixing length defined in models (2) and (5) is a maximum length scale that limits the dissipation of energy, ensuring
that dissipation in the models is not underrepresented (Isfahani and Brethour, 2009).

Models (6) and (7) function differently from the RANS eddy viscosity models. The accuracy of the LES model, model (6), depends on knowledge of the flow conditions so that the filter scale can be defined to allow for most of the large-scale turbulence to be resolved within the model itself rather than in the sub-grid representation of the small-scale turbulence (Pope, 2000). The results from model (6) should be viewed considering these limitations, since the grid size was not determined
according to the scale of turbulence that should be resolved in the model. Model (7) ignores turbulence and simulates the flow as entirely laminar. As turbulence tends to dissipate energy, this model will under-represent dissipation.

### 2.1.2 Sensitivity to Grid Size

Model results tend to improve with grid refinement. The grid cell size used for the simulations was selected to allow for sufficient resolution of the topographic and bathymetric features of the glacial lake as well as the dynamic wave features during
the wave generation and overtopping phases while also balancing time and computational resources. To assess the impact of grid size on model results, a simulation was run with a coarser grid.

The regular mesh used in the FLOW 3D model consists of 6 m x 5.33 m x 6.5 m grid cells in the x-, y- and z-directions, respectively, spanning distances of 2400 m (x-direction), 800 m (y-direction), and 650 m (z-direction). For the grid size sensitivity analysis, a coarse grid simulation, with double the original cell grid size, was run for the large avalanche source
scenario at the current lake level.

For the coarse grid simulations, the same value for water depth at each time step in the coarse grid was assigned to the four smaller cells that fall within each cell in the coarser grid to allow for direct comparison between the results of the coarse grid simulation and the regular mesh. To compare the coarse grid results to the results from the regular model mesh, the root-mean-square error (RMSE) of fluid depth for all grid cells within the lake was calculated at each time step. Additionally, the percent
difference in peak overtopping flow rate and total overtopping volume and the RMSE of the outflow hydrograph were calculated for the coarse grid simulation.

**2.2 Boundary Conditions: Representing Avalanche Impact**

The problem of reproducing an avalanche-generated impulse wave in a hydrodynamic model of a glacial lake presents a challenge because of the complicated dynamics of mixing and dissipation of energy that occur at the point of impact; these processes are difficult to represent correctly in the model. The results of avalanche simulations performed in the Rapid Mass Movements (RAMMS) model (Christen et al., 2010; Bartelt et al., 2013), reported in Somos-Valenzuela et al. (2016), were used to generate inputs to the lake model. Two different methods of representing the impact of the avalanche with the lake and the corresponding mass and momentum transfer were tested to determine the sensitivity of the lake model to the boundary conditions. The variability in the results between the two boundary condition methods gives an approximation of the uncertainty associated with the avalanche impact and wave generation.

**2.2.1 Avalanche Source**

The avalanche source boundary condition method represents the avalanche entering the lake by simulating water flowing from the lower glacier slopes into the lake. The density of the avalanche material that is typical for this type of GLOF, the mixture of snow, rock and ice, is nearly the density of water (Schneider et al., 2014); therefore, water was used in place of the avalanche fluid, and the volume of the water that represents the avalanche was the same as the total avalanche volume. This is the same approach used by Worni et al. (2014) and Fah (2005). The two fluids (water and the avalanche material) have different viscosities, but the model was adjusted to account for the effects of the lower viscosity of water (less dissipation of energy as it flows towards the lake). The depths and velocities of the avalanche entering the lake from the RAMMS model lake were matched in the FLOW 3D model by varying the height at which the initial avalanche fluid volume was released above the lake and the initial depth of the avalanche fluid in the FLOW 3D model. The momentum transfer from the avalanche to the lake is what generates the displacement wave. Thus, the wave characteristics depend both on the mass (equivalent to depth) and velocity of the avalanche as it enters the lake. If the mass and momentum of the flow representing the avalanche impacting the lake are similar in the FLOW 3D avalanche source model and RAMMS, then the FLOW 3D simulations should realistically represent the wave generation. Reflected waves may be somewhat different due to the potential settling of the avalanche material that cannot be represented in the FLOW 3D model, but these differences are probably minimal because the magnitude of the reflected wave is much less than the initial wave.

**2.2.1 Mass-momentum Source**

The second boundary condition method for representing an avalanche impacting the lake was a mass-momentum source. For this method, hydrographs were constructed from the RAMMS avalanche simulations approximating the volumetric flow rate of the avalanche entering the lake by taking the depth and velocity from RAMMS at various points (approximately 10-15 points) along the edge of the lake for each time step. The average avalanche depth, velocity, and flow rate were calculated for each time step. These avalanche hydrographs were slightly altered so that the total volume was equivalent to the avalanche

volume, and the resulting adjusted hydrographs were used as the inflow boundary condition of the FLOW 3D model, representing the input of mass and momentum that generates the impulse wave. This was done using the mass-momentum source function in FLOW 3D with the boundary condition defined by the hydrograph and cross-sectional area of the flow entering the lake.

## 2.3 Wave Characteristics

There are five main phases of an avalanche-generated impulse wave in a glacial lake: (1) wave generation from the avalanche entering the lake, (2) propagation of the wave across the lake, (3) run-up on the damming-moraine, (4) overtopping of the moraine, and (5) reflected wave(s) from the portion of the wave that does not overtop the moraine. The characterization of these phases of an avalanche-generated wave is important because empirical methods (e.g., Heller and Hager, 2010) have been developed to model wave generation, but wave propagation often cannot be accurately described by simple empirical equations, especially for glacial lakes with varying bathymetry. Wave generation is dependent primarily on the avalanche characteristics and the lake depth at the point of impact; whereas, wave propagation is dependent on initial wave characteristics, lake bathymetry and the surrounding topography.

The primary parameters used to study the wave characteristics were the maximum height of the wave in the lake and the wave height as it overtopped the moraine dam. The maximum wave height, as a function of distance along the lake, was calculated to assess how the wave changes during the propagation phase and to allow for comparison with the empirical method of Heller and Hager (2010). At this point, the difficulty of model validation and uncertainty quantification must be mentioned. In this work, events are modeled that have not yet occurred, and very little data are available from similar past events that can be used to calibrate or validate model results. There was a landslide at Lake Palcacocha in 2003 that overtopped the lake-damming moraine and caused some damage to the structure of the moraine complex. The approximate volume of this landslide is known, and the wave height was estimated to be 8 m based on the fact that moraine overtopping occurred (Vilimek et al., 2005). However, it is possible that the actual wave height in the lake was less than the estimated value because the wave height generally increases during run-up on the damming moraine. The information available for the 2003 landslide is insufficient for validation of this lake model. Similarly, the 2010 GLOF that occurred at Lake 513 in the Cordillera Blanca of Peru provides some information to compare results to, but that event occurred at a lake with unique characteristics (solid rock damming-moraine) and there is some discrepancy among the estimates of the avalanche magnitude, wave height and overtopping volume (Carey et al., 2012; Valderrama and Vilca, 2012; Schneider et al., 2014). Therefore, the results of the empirical model (Heller and Hager, 2010) were used to compare with the FLOW 3D hydrodynamic lake modeling.

The empirical method of Heller and Hager (2010) for calculating characteristics of impulse waves is based on a database of field measurements and laboratory experiments. If the characteristics of the impulse wave in both the hydrodynamic and empirical models are similar, then there is reason for confidence in the hydrodynamic model results. However, the empirical method is only an approximation based on simplified representations of lake geometry and avalanche characteristics. The method has certain acceptable ranges of variables, such as relative slide density, volume, width, and Froude number, for which

the empirical equations hold true. For Lake Palcacocha, all the variables fall within the acceptable ranges except the relative slide width; therefore, the wave characteristics calculated according to this method can be reasonably relied upon to compare with the three-dimensional simulation results, but only to get an idea of the approximate wave dimensions.

### 2.4 Scenarios

Two sets of scenarios were simulated with the hydrodynamic model: avalanche scenarios and lake-lowering scenarios. To assess the current GLOF hazard, simulations were first run with the current lake level (the baseline level). The baseline level was defined as the lake level controlled by the current outlet works, a tunnel that maintains a freeboard level of 8 m and a water surface elevation of 4562 m. Three avalanche scenarios were used to represent a range of potential avalanche sizes that might impact the lake: small ($0.5x10^6$ m$^3$), medium ($1x10^6$ m$^3$) and large ($3x10^6$ m$^3$). The avalanche characteristics for each

scenario are given in Somos-Valenzuela et al. (2016). Second, scenarios with different lake levels were simulated to study how lowering the lake surface might influence the overtopping wave volume and discharge. These scenarios included lowering the lake level 15 m and 30 m from the baseline lake level and were selected based on what has been proposed by local government technical specialists in Huaraz as plausible lake risk mitigation strategies.

Each lake level scenario (including the baseline) was simulated for all three avalanche scenarios, forming a total of 9 scenarios;

the overtopping volume and outflow hydrograph were computed for each scenario. Lake lowering scenarios were analyzed for reduction in peak overtopping flow rate and total discharge volume. Although the goal of this work is examining lake hydrodynamics, the greater aim is to assess the potential for GLOFs to impact downstream populations. Simulations of downstream inundation and flood intensities can facilitate analysis of lake lowering schemes to reduce GLOF hazard levels. Somos-Valenzuela et al. (2016) evaluated how lake lowering may alter the GLOF impacts in Huaraz for the avalanche source

scenarios and found that overtopping volumes of 20,000 m$^3$ or less would not result in significant flooding in Huaraz. Considering this, three classifications were used to describe the overtopping results for each scenario and their potential downstream impacts: (a) no discharge, (b) medium discharge $\leq 20x10^3$ m$^3$, (c) and high discharge $> 20x10^3$ m$^3$. Classification (a) implies that there should not be any downstream impacts. For scenarios resulting in medium discharge, classification (b), the downstream impacts should be minimal, and scenarios resulting in high overtopping discharges, classification (c), there

could be significant downstream impacts. However, these classifications should be considered in light of the significant uncertainty in the overtopping estimates. A comprehensive probabilistic hazard assessment and evaluation of mitigation alternatives is beyond the scope of this work, and these classifications of the magnitude of overtopping discharges are only intended to provide a useful tool that can be used in the decision-making process.

## 3 Results

For each scenario, FLOW 3D was used to model the avalanche-generated impulse wave, from the wave generation to the overtopping phases. For each avalanche event, simulations were run using both boundary condition methods (avalanche and mass-momentum sources), first for the baseline lake level and then for the two lake-lowering scenarios.

### 3.1 Sensitivity Analysis: Turbulence Model and Grid Size

#### 3.1.1 Sensitivity to Turbulence Model

For the large size scenario with the avalanche-source boundary condition and current lake level, the results of using the various turbulence models were compared to the baseline model (1) (RNG-dynamic mixing length). The RMSD (Fig. 3) shows the average difference in fluid depth between the baseline model and each of the other turbulence models. For all models, the highest RMSD values were for times up to 50 s when the water surface is most actively changing as the impulse wave is generated and begins to propagate across the lake. Models (6) (LES) and (7) (laminar) show the most deviation from the baseline model with maximum RMSD values around 2.5 m. The laminar model shows high deviations from the baseline model because it does not account for turbulence and should be the least dissipative of all the models. This is reflected in the peak flow rate, overtopping volume and maximum wave height (Table 1), which were all higher than the baseline model. The LES model appears to be overly dissipative, giving the lowest values for all parameters used for comparison between the models. It is difficult to say why this is the case, but it could be due to inhomogeneity in the flow or numerical errors due to the filter scale.

Models (2) (RNG-constant mixing length), (3) (k-epsilon) and (4) (Prandtl mixing length) may be more appropriate for this type of simulation. The results from these models more closely align with the baseline model; however, there are still differences in fluid depth between the models. All three models had maximum RMSD values for fluid depth of around 1.8 m; the models approached a steady state (RMSD of approximately 0.5 m) after 200 s when the initial wave overtopped the moraine. The highly variable lake bathymetry and fluid depths make defining an appropriate mixing length difficult and introduce a source of uncertainty in the model; many of the turbulence models require the definition of a mixing length that ensures that the dissipation of energy is not underrepresented in the model. For this reason, model (1) appears to be the optimal choice in this case. Yet, the similarity in the results between the RANS eddy viscosity models (1) − (5) indicates that the uncertainty introduced by the constant mixing length models is relatively insignificant.

The RMSD of the overtopping hydrograph flow rates for each of the turbulence models are given in Table 1 along with additional comparisons of the hydrographs, including the percent difference in peak flow rate and total overtopping volume. The largest differences in flow rate and overtopping volume came from models (6) (LES) and (7) (laminar) with the laminar model producing higher flows and the LES model producing the lowest flow rates. The hydrographs from the other models resembled that of the baseline model. The percent differences in peak flow rate from the eddy viscosity models ranged from around 0.25% for model (4) (Prandtl) to around 3% for model (3) (k-epsilon). The differences in total overtopping volume

were a little higher, although all were less than 5%, and the differences in maximum wave height were much less significant for all but model (6) (LES), with most models giving differences less than 2%.

The laminar model (7) is the only model that gave higher flow rates and overtopping volumes than the baseline model, indicating that even if the turbulence model introduces uncertainty into the model results, the results of the baseline model are most likely conservative, giving possibly higher discharges. Considering all the other sources of uncertainty in the models of the avalanche and wave generation, the turbulence model is one of the less significant sources of uncertainty.

### 3.1.2 Sensitivity to Grid Size

The RMSE of fluid depth for the coarse grid simulation compared to the regular mesh is a good measure of the error introduced by changing the grid resolution (Fig. 4). The highest errors were in the first 50 s of the simulation time, during the wave generation, propagation and run-up phases. However, there was a baseline level of error that comes simply from extrapolating the initial conditions to a coarser grid because the bathymetry and initial fluid depths are better represented in the fine grid model; this baseline error was unavoidable because the resolution of the bathymetry must be the same as the resolution of the model grid, so we lose some of the precision of the bathymetric representation in the model with the coarse grid. The RMSE at $t = 0$ reflects this error. After 50 s, the RMSE began to level off at a relatively consistent level of approximately 1.5 m. This was about three times higher than the RMSD from the eddy viscosity turbulence models at the same point in time, indicating that grid size could introduce much more error than the turbulence model.

The RMSE of overtopping discharge for the coarse grid simulation was approximately 3300 m³/s. This amount of error is not insignificant; it is approximately three times the RMSD for the eddy viscosity turbulence models but less than the RMSD for the laminar flow model. The peak discharge from the coarse grid simulation was over 5% higher than the peak discharge from the regular grid size model (a difference of 4,200 m³/s). The total overtopping volume was slightly higher for the coarse grid simulation (a difference of 30,000 m³), but the difference was less than 1%, so the coarse grid model seems to estimate the total overtopping volume well even if it does not get the wave dynamics and outflow hydrograph completely correct. Although the error resulting from using a coarser mesh was greater than the uncertainty from most of the turbulence models, the uncertainty due to the grid size is still not a very large source of error.

### 3.2 Comparison of Boundary Conditions: Avalanche Source vs. Mass-momentum Source

The inflow hydrographs of the two boundary condition methods are shown in Fig. 5 along with the hydrograph from the RAMMS avalanche model (Somos-Valenzuela et al., 2016). For all three avalanche scenarios, the peak inflow for the avalanche source was significantly higher than for the mass-momentum source. The mass-momentum boundary condition inflows were very close to those of the RAMMS model in each case because the boundary condition was defined to match the RAMMS avalanche hydrograph. The higher peak inflows for the avalanche boundary condition are probably because the lower viscosity of water relative to the avalanche material allows the fluid to flow and spread out more quickly; to compensate for this, the avalanche boundary condition fluid release volume was concentrated over a smaller area so that the fluid depths would

not be too low, but the result was higher inflow rates over a shorter period. The peak inflow rates for the avalanche boundary condition ranged from nearly twice the peak flow rate of the RAMMS avalanche for the large scenario to over 5 times higher for the small scenario, but the inflows for the avalanche boundary condition were of much shorter duration. For the large scenario, peak overtopping discharge for the mass-momentum boundary condition (Table 2) was 14% less than the discharge for the avalanche boundary condition (compared to a difference of about 50% for the inflows). However, for the medium and small scenarios, the difference in peak overtopping discharge between the two boundary condition models was more pronounced. For the medium mass-momentum boundary condition, the overtopping discharge was 65% less than the discharge from the medium avalanche boundary condition; this difference was only slightly lower than the difference in peak inflow (~75%). The overtopping discharge for the small mass-momentum boundary condition was almost 91% less than the discharge for the small avalanche boundary condition (with difference in peak inflow of around 80%). While the difference in overtopping volumes for the *large* avalanche and mass-momentum boundary condition was only 9%, the total overtopping volume for the *small* mass-momentum boundary condition was over an order of magnitude less than the overtopping volume resulting from the small avalanche boundary condition (Table 2).

There were a few irregularities in the inflow hydrographs that should be mentioned. First, the large avalanche source inflow hydrograph had a bimodal peak, likely due to the way in which the initial avalanche fluid volume was defined. The initial fluid volume was defined as blocks of water above the natural terrain, the surface elevations of which were set at graduated levels, taking the shape of steps to more closely mimic the natural descent of the terrain and have a relatively constant initial water depth; this definition of the initial fluid release volume is not realistic, but after it was released, the fluid flowed into a more natural state. However, for the large avalanche source, the sections of the initial fluid volume most likely had variations in the initial water surface elevation that were too abrupt so that the fluid did not coalesce into one continuous surface but rather had two areas of peak flow depth. This is a problem that results from releasing blocks of water just above the lake; the initial fluid volume is not realistic, but water will even out into a natural flow before it reaches the lake. The fluid cannot be released at a point that is too high or the velocities will be excessive, but to get a high enough volume with accurate depths, it is difficult to get an even flow by the time the water reaches the lake. A second irregularity was the smaller, second peak in the inflow hydrographs from the avalanche boundary condition in the medium and small scenarios, likely the result of flow entering the lake from the sides. This is not unrealistic, since there was inflow from the sides of the lake in the avalanche model. However, due to the higher viscosity of the snow-rock-ice mixture of the avalanche, the inflow from the lateral moraines probably would happen more gradually so that the abrupt inflow from the sides would not cause such a significant peak in the inflow hydrograph.

**3.3 Wave Characteristics**

The impact of the avalanche with the lake generates a large displacement wave. As the wave propagates across the lake, it reaches a maximum height as it approaches the shallow part of the lake near the damming-moraine (Fig. 6). The characteristics of the waves generated for each avalanche scenario are given in Table 3. The FLOW 3D wave heights were all larger than the

empirically-calculated wave heights (Heller and Hager, 2010); however, the waves were of a similar magnitude with both methods with a difference in maximum wave height between FLOW 3D and the empirical method of 14% (5.8 m) for the large avalanche source. The FLOW 3D results showed attenuation of the wave as it propagated along the lake; this attenuation resulted in a reduction in the wave height of approximately 30% before the wave began the run-up phase (Fig. 6).

Upon closer examination, the wave generated from the large avalanche source (Fig. 6) had two peaks that were of similar height. The first peak was near the avalanche impact, corresponding to the location of the wave represented by the empirical equations; the second peak, that was slightly higher, occurred as the wave began to run up on the shallower part of the lake. The wave characteristics calculated by the empirical method consider the wave generation process but do not account for the impact of run-up on the wave characteristics. Therefore, the peak wave height in the deeper portion of the lake is the closest

point of comparison with the empirical equations. Fig. 6 gives the wave height as a function of distance along the lake (not as a function of time); there were some oscillations in the profile of the maximum wave height, most likely due to splashing from the run-up on the sides that was reflected off the lateral moraines and returned to the lake at irregular intervals.

### 3.4 Overtopping Hydrographs and Volumes

The run-up phase culminates with the moraine overtopping; the wave heights given in Table 2 correspond to the height above

the moraine crest as the wave overtops the damming-moraine. The volume of water that resulted from the overtopping of the moraine was significant; the total overtopping discharge volume for each scenario is given in Table 2, and the overtopping hydrographs are shown in Fig. 7. The large avalanche source resulted in a peak overtopping discharge of approximately 63,000 $m^3$/s that occurred around 60 s after the start of the avalanche as well as a smaller peak of 6,000 $m^3$/s resulting from the overtopping of the reflected wave. The overtopping of the initial wave lasted about 100 seconds for the large avalanche source,

70 seconds for the medium avalanche source, and 50 seconds for the small avalanche source.

The mass-momentum boundary condition consistently resulted in lower overtopping discharges and volumes, but the differences between the mass-momentum and avalanche boundary condition were more pronounced for the small and medium scenarios. For the large mass-momentum boundary condition, the peak overtopping flow rate was 14% less than that of the avalanche boundary condition. The large mass-momentum boundary condition overtopping volume was 11% less than the

avalanche boundary condition overtopping volume. For the medium mass-momentum boundary condition, the peak discharge and overtopping volume were 65% and 70% less than the avalanche boundary condition, respectively, and the difference in both the peak discharge and overtopping volume between the small avalanche and mass-momentum boundary conditions was 91%.

The overtopping volumes for all scenarios were less than the volume of avalanche material entering the lake. The overtopping

volume for the large avalanche boundary condition was 60% of the avalanche volume, and for the medium and small avalanche boundary conditions, the overtopping volumes were 50% and 30% of the avalanche volumes respectively. The overtopping volume decreases relative to the avalanche volume as the avalanche size decreases, indicating that the lake has more capacity to dissipate smaller avalanche-generated waves.

**3.5 Lake Lowering Scenarios**

Two scenarios of lake lowering were simulated to evaluate the potential effect of lowering the lake level as a mitigation strategy. Three avalanche sizes and both types of boundary conditions were simulated with each lake level, resulting in a total of 18 simulations. The overtopping volumes and peak discharges were somewhat reduced by lowering the lake 15 m, while 30 m lowering resulted in even further reductions in overtopping discharges (Table 2). The hydrographs for the overtopping discharge are shown in Fig. 8.

Lowering the lake level, even by as much as 30 m, did not completely prevent overtopping of the damming-moraine. Nonetheless, overtopping may be prevented by lake lowering for smaller avalanches. A 90% reduction of overtopping volume may be achieved for the medium avalanche boundary condition through lowering the lake level by 30 m. Overtopping was not avoided entirely with the 15 m lake lowering, but the overtopping volumes and discharges were approximately 60% and 80% less than with the current lake level for the medium and small avalanches, respectively. Lake lowering appears to have the least impact for large avalanches, as significant overtopping still occurred under all lake lowering scenarios for a large avalanche. However, the overtopping volume was reduced by 28% for the large avalanche boundary condition, with 30 m lake lowering and by 73% for the large mass-momentum boundary condition, with 30 m lake lowering. The categorization of each scenario according to the overtopping volume (Sect. 2.4) is given in Table 4.

The overtopping wave heights increased with lake lowering even though the total overtopping volumes and peak flow rates decreased. This may seem counterintuitive, but it can be explained by looking at how the lake dynamics may be expected to change with lake lowering. First, as the water surface level is lowered, the total volume stored in the lake decreases, thus the momentum transferred to the lake from the avalanches per unit volume should be higher. The total volume in the lake decreases with lake lowering, so the additional momentum relative to the lake volume can produce taller waves. Secondly, as the point of avalanche impact is at a lower elevation relative to the avalanche release area with lake-lowering, there is more momentum in the avalanche fluid when it enters the lake. Although the increased overtopping wave heights for the lake lowering scenarios indicate that the waves may be larger when the lake is lowered, the amount of overtopping still decreases with lake lowering. This is most likely due to the lower initial water surface elevation; the lower free surface elevation results in a larger freeboard and means that more momentum is required for overtopping; although the momentum transfer per unit volume to the lake from the avalanche is greater, more of this momentum is lost during the run-up and overtopping, and less water is actually able to pass over the crest of the terminal moraine.

# 4 Discussion

## 4.1 Boundary Condition Methods

Although the avalanche boundary condition seems to have more uncertainty than the mass-momentum boundary condition, each boundary condition method has its limitations. The complex nature of the interacting dynamic physical systems makes it

difficult to develop a comprehensive and precise method for simulating avalanche-generated waves in glacial lakes. Recent advances in two-phase flow models such as r.avaflow (Pudasaini, 2012; Mergili, 2017) can facilitate simulations of wave generation from avalanches entering a lake; however, the use of depth-averaged equations still limits the ability to use this type of model to simulate all of the phases of an avalanche-generated wave from wave generation to overtopping. Avalanches
typically consist of a mixture of snow, ice and rock, and the biggest limitation of the boundary conditions in this model is the representation of the avalanche fluid as water because the dissipation of energy of the actual avalanche material is different from water. This limitation can be partially overcome by calibrating the model to replicate the depth and velocity characteristics of the avalanche as it enters the lake. This is done by adjusting the avalanche release area in the avalanche boundary condition and the hydrograph and cross-sectional area of the inflow for the mass-momentum boundary condition. However, it is
impossible to completely replicate the avalanche characteristics in the lake hydrodynamic model, and there are significant differences in the inflow hydrographs of the FLOW 3D model and the RAMMS avalanche model (like the mass-momentum source) when the avalanche boundary condition is used. The discrepancies between the avalanche and mass-momentum boundary conditions are more pronounced for smaller avalanches, but there is no obvious solution to overcome this difficulty when using the avalanche boundary condition. To further advance the simulation of avalanche-generated waves, models are
needed that can easily and accurately represent two distinct fluids (in this case the mixture of snow, rock and ice of the avalanche and the water in the lake) combined with non-hydrostatic free surface flows. Without two-phase models that can simulate free surface flows, it will not be easy to overcome the limitations and irregularities of the model that result from the representation of the avalanche fluid as water.

The avalanche boundary condition has much higher and possibly unrealistic peak inflow rates, but it gives a better physical
representation of the actual geometry of the terrain as the avalanche enters the lake. The avalanche boundary condition is also able to simulate the effects of avalanche material entering the sides of lake, whereas the mass-momentum boundary condition only simulates flow entering from the end of the lake. The mass-momentum boundary condition better matches the peak flow rates of the avalanche because that is how the method was designed; the flow rate of the avalanche inflow is a control parameter for the mass-momentum boundary condition. However, under this boundary condition, the avalanche material enters the lake
horizontally, rather than on the steep incline of the actual terrain above the lake. Therefore, this boundary condition likely underestimates the momentum transfer between the avalanche and the lake, as the avalanche can gain more momentum as it enters the lake at a downward angle. Despite the limitations of each boundary condition method, they are representing a range of possible outcomes, and the results could be considered as upper and lower bounds on the overtopping discharge from the lake model. Because we do not have any field measurements of the characteristics of avalanche-generated waves during GLOF
events or the resulting discharge hydrographs, we do not possess the means of validating the model results presented in this paper or conclusively evaluating the boundary condition methods.

The avalanche simulation is the process in the GLOF chain of events that carries the greatest uncertainty because avalanche dynamics may be the least understood of the processes. The range of uncertainty in the avalanche conditions (depths, flow rates and velocities) is possibly greater than the range of variability in the inflow hydrographs for the lake model. We have no

estimates of the uncertainty in the avalanche model, but any uncertainties in the avalanche simulations are propagated into the lake model and subsequent processes in the GLOF chain of events. Although there is significant variability between the avalanche and mass-momentum boundary condition results, the range of variability in the peak flow and shape of the avalanche hydrographs may be even greater than the variability in the discharge hydrographs from the lake model.

## 4.2 Wave Characteristics

The characteristics of the wave as it propagates across the lake are significant indicators of the magnitude of the event that is being simulated. The wave heights are quite large (up to nearly 50 m tall) when compared with the initial depths of the lake that range from 72 m to less than 10 m (UGRH, 2009). Such large waves relative to the lake depths indicate that vertical

accelerations are significant and should not be neglected. Thus, a non-hydrostatic model is essential for accurately representing the wave dynamics. Because the type field data that would be needed for model validation (e.g., wave characteristics such as wave height and attenuation) were not available, wave heights from the FLOW 3D simulations were compared with those calculated with the empirical equations of Heller and Hager (2010). The empirical model has been compared to measured data and laboratory experiments (a form of validation of the method), so it may reasonably be concluded that if the 3D model gives

similar values for the wave characteristics, we can have more confidence in the 3D model. However, this comparison is only valid for the wave generation phase and maximum wave heights, as the empirical model does not represent the wave propagation, run-up and overtopping phases well. The FLOW 3D simulations did a reasonable job reproducing the maximum wave heights, especially for the larger avalanche scenarios. The FLOW 3D simulations also account for lake bathymetry and give a more accurate representation of the dissipation of energy during the propagation phase; thus, the FLOW 3D model can

likely produce more realistic overtopping discharges than the empirical method. For the large avalanche scenario, both boundary conditions resulted in wave heights that were only 4.4-5.8 m higher than the empirically calculated wave heights. However, it is worth noting that the maximum wave height for the large avalanche boundary condition occurred at the beginning of the run-up phase in the shallow part of the lake, and the first wave peak in the deep portion of the lake was closer to the empirically predicted height. The large differences between the empirical and FLOW 3D wave heights for the medium

and small scenarios may be due to the shortcomings of the avalanche boundary condition. Nevertheless, the relatively close agreement between the empirical and hydrodynamic models for the large avalanche scenario indicates that it may be possible to use the empirical method as a calibration tool.

During the run-up phase of the wave propagation, two things happened simultaneously. The wave height increased due to the run-up in the shallow portion of the lake, but there was also some energy loss due to the sharp discontinuity in the lakebed

geometry. Generally, one might expect the wave height to increase even more than what occurred in the FLOW 3D simulations; however, due to the lakebed geometry, there is more dissipation of energy when the wave reaches the shallow portion of the lake than would occur if there were a more gradual transition between the deep and shallow areas of the lake.

### 4.3 Model Sensitivities and Uncertainties

When model calibration and validation with field data is impossible, it is important to understand the model sensitivity to input parameters. This sensitivity analysis may be used in lieu of validation in order to better understand the uncertainties of the model so that we do not represent more confidence in the model results than is justifiable given the uncertainties in the modeling process. The greatest uncertainty in the lake modeling arises from the wave generation and avalanche characteristics. Uncertainties due to the turbulence model and grid size are not negligible, but they are small compared to the magnitude of uncertainty from the wave generation.

The input parameter that seems to generate the least sensitivity is the turbulence model, with most variables used to evaluate the sensitivity varying less than 3%. Although the results were somewhat more sensitive when the LES model and laminar flow model were used, this may be expected because neither model would be considered an appropriate choice for this application. We have insufficient information to correctly apply the LES model, and we know that the flow is not laminar, so neglecting turbulence altogether would underrepresent the dissipation of energy in the model. The RANS turbulence models all gave similar results. Therefore, the choice of turbulence model should likely have little impact on the results of the lake model, even when input parameters such as the mixing length are unknown. Nonetheless, the turbulence models that use a dynamically computed mixing length are still probably the best choice because they eliminate the need for assumptions about the flow characteristics.

While the model sensitivity to grid size is greater than the sensitivity to the turbulence model, the variability in the analysis parameters was still generally less than 5%. In addition, the coarse grid simulation gave a larger overtopping volume and peak flow, so even though some error may be introduced by increasing the grid spacing, in this case it gave a more conservative result. However, any conclusions made from these results should be done carefully. The analysis of the effect of grid size on model results presented in this paper is not comprehensive, and it may be that further increasing the size of the grid cells could have an undesirable effect on the reliability of the model results. Nonetheless, the choice of grid size is likely to be a much less significant source of uncertainty than the boundary conditions.

One way to estimate the uncertainty in the wave generation is by using more than one method to represent the impact of the avalanche with the lake (i.e., the two methods for modeling the boundary conditions). Without any in situ data from real events, the level of uncertainty cannot be estimated precisely, but given the range of overtopping flows and volumes from the two boundary condition methods, the uncertainty is considerable. Although there is no way to validate the results to know which type of boundary condition is more representative of the actual conditions, it is possible that the avalanche boundary condition is overestimating the momentum transfer while the mass-momentum boundary condition is likely underestimating it. The avalanche boundary condition could represent an upper bound for the simulation results while the mass-momentum source may be closer to a lower bound.

The scenarios of avalanche size cover a range of possible avalanche volumes that could trigger GLOFs of significant size, and the analysis of the results from these scenarios may be considered as a measure of the sensitivity of the overtopping discharge

to the input parameters related to avalanche characteristics. The avalanche characteristics carry their own uncertainties that are propagated into the lake model and subsequent downstream processes, and these uncertainties are probably much greater than the uncertainties associated with the turbulence model and grid size. Although we do not have enough information to quantify the uncertainty from the avalanche model, evaluating the potential effect of different sized avalanches on overtopping
discharges can help us better understand the downstream sensitivity to the avalanche characteristics.

## 4.4 Implications of Model Results

This paper focuses exclusively on the lake hydrodynamics and does not consider the uncertainties in the avalanche simulations or the question of dynamic erosion of the terminal moraine due to overtopping flows. The avalanche is the portion of the GLOF
process chain that is the least understood and most likely the greatest source of uncertainty in GLOF modeling and hazard mapping. Although the uncertainty resulting from the avalanche portion of the chain of events must be considered in the decision-making process, investigating the uncertainty in the avalanche characteristics is beyond the scope of this work. In a way, the avalanche scenarios are attempting to capture some of that uncertainty, but it does not represent all of the uncertainty associated with the avalanche model. This work explores the uncertainties that arise when representing the impact of the
avalanche with the lake and the wave generation, but this is necessarily limited by the avalanche characteristics that were available from the avalanche simulations. Until further advances are made in the field of avalanche simulations and we gain an improved understanding of avalanche dynamics for this type of event, it is impossible to incorporate all of the uncertainty of potential avalanches into analysis of the dynamics of avalanche-generated waves. All that we can do is assess the sensitivity of the lake model to different types of inputs to gain a qualitative understanding of how these uncertainties might impact the
characteristics of avalanche-generated waves and the overtopping discharges. The potential erosion of the terminal moraine is also an important factor to consider when assessing the hazard level of any glacial lake with a moraine dam. For Lake Palcacocha, this was assessed by Somos-Valenzuela et al. (2016) through a separate hydromorphodynamic model, and the conclusion was that despite significant potential for erosion, the moraine is extremely unlikely to fail.

The results from the large avalanche simulations represent the worst-case scenario of an avalanche-induced GLOF from Lake
Palcacocha if the moraine is as stable as it seems. Given the significant differences between the small and medium avalanche simulations, results from both boundary condition methods should be provided if these scenarios and their likelihoods will be used in an economic or risk and vulnerability analysis of the mitigation alternatives. All the large avalanche scenarios and most of the medium avalanche scenarios resulted in significant overtopping, even with lake lowering. Thus, it is clear that steps to mitigate or reduce the hazard must be taken because even with the low end of the range of uncertainty in avalanche sizes, the
resulting discharges could represent an unacceptable level of risk for the city of Huaraz, as was also indicated in Somos-Valenzuela et al. (2016). However, the classification of overtopping discharges by volume used here (Table 4) is not fully indicative of the effect of lake lowering on hazard mitigation. The downstream impacts for each scenario should be considered when evaluating lake lowering scenarios, but decision makers must also recognize the uncertainty contained in these GLOF

hazard assessments. The potential for lake lowering works to prevent overtopping for the small and medium avalanche scenarios is significant because small and medium avalanches are believed to be much more likely than large avalanches (Huggel et al., 2004); therefore, the real impact of lake lowering may be more than is immediately apparent with these results. However, from the modeling results alone it is not possible to determine an optimum lake level. Further economic and vulnerability analyses are necessary to recommend an ideal mitigation alternative.

**5 Conclusions**

Three-dimensional non-hydrostatic models can be a useful tool to simulate avalanche-generated waves and improve our understanding of lake dynamics during GLOF events. The simulations of Lake Palcacocha show that waves of considerable magnitude can be produced. While sensitivity of the overtopping discharge to the turbulence model and grid size is minimal, the avalanche characteristics and the shape of the inflow hydrographs substantially influence the overtopping wave volumes. While large avalanches produce the largest overtopping discharges, even smaller avalanches could generate significant overtopping discharges. Based on the downstream inundation analysis in Somos-Valenzuela et al. (2016), even the small avalanche scenario could result in substantial inundation in the city. Somos-Valenzuela et al. (2016) only evaluate scenarios that use the avalanche source boundary condition, but the results presented here indicate that the overtopping discharges with the mass-momentum source boundary condition may be lower than those evaluated for downstream impacts in Somos-Valenzuela et al. (2016). Based on this, it can be concluded that there is a considerable amount of uncertainty in the lake model due to the boundary condition method. However, even considering this uncertainty, we can still conclude that overtopping discharges for the current lake level could be significant. Lowering the lake level may reduce the overtopping volume and discharge for a large avalanche, but it is not possible to eliminate the potential for overtopping. For small ($0.5 \times 10^6$ m$^3$) and medium ($1 \times 10^6$ m$^3$) avalanches, it may be possible for the wave to be contained in the lake if the water surface is lowered. However, given the range of uncertainty in the model results, it cannot be stated conclusively that lowering the lake level would prevent overtopping for smaller avalanches. Even though the precise reduction in hazard level due to lake lowering cannot be quantified using the given approach, it is reasonable to conclude that lowering the level of Lake Palcacocha can reduce the hazard levels in the city of Huaraz.

The modeling reported here provides a significant advancement beyond previous simulations of avalanche-generated waves. Model calibration is less important for the three-dimensional modeling approach due to the improved representation of physical processes as compared with two-dimensional SWE models; therefore, it presents an alternative that can be used when field data from a prior GLOF are not available for model calibration. Despite the advantages of this method, uncertainties are still present; however, as the fundamental physical phenomena are better represented in three-dimensional models, errors can be attributed more to uncertainties in the physical parameters, initial and boundary conditions rather than the model constructs. Nonetheless, the lake dynamics still remain a problematic link in attempts to model the GLOF process chain. The sensitivity analyses presented in this paper should help future modelers understand the uncertainties associated with the modeling of

displacement waves and assist them in determining which input parameters need the most attention. Given the considerable sensitivity of the lake model to the boundary condition method (representation of the impact of the avalanche with the lake), it is recommended that more than one boundary condition method be used. Until we gain a better understanding of the dynamics of mass movements and their influence on wave generation, it is best to consider a range of possible outcomes rather than selecting just one method and assuming that it accurately depicts the wave characteristics.

Avalanche simulation is the GLOF process chain link that carries the greatest uncertainty, and much of that is propagated into the lake model. Precise knowledge of avalanche behavior is limited, and so it is difficult to evaluate how well the lake model represents the avalanche as it enters the lake. Because the lake model is so heavily influenced by the avalanche characteristics, it is hard to quantify the uncertainty in the wave simulations. More studies are needed to gain a better understanding of the magnitudes and sources of uncertainty in glacial lake modeling of waves generated by mass movements.

## Competing Interests

The authors declare that they have no conflict of interest.

## Acknowledgements

The USAID Climate Change Resilient Development (CCRD) and Sustaining Mountain and Water Livelihoods (SMWL) projects have provided support that made this work possible. The authors would like to thank the software developers from Flow Science, Inc. for the FLOW 3D license and technical assistance. Marcelo Somos-Valenzuela, Denny Rivas, and Cesar Portocarrero were indispensable resources who gave helpful feedback and encouragement. The authors are very grateful for the constructive comments of the reviewers.

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

**Table 1. Comparison of overtopping hydrograph characteristics among turbulence models.**

| # | Model | RMSD ($m^3/s$) | Difference in Peak Flow Rate | | Difference in Overtopping Volume | | Difference in Maximum Wave Height | |
|---|---|---|---|---|---|---|---|---|
| | | | $m^3/s$ | % | $10^6\ m^3$ | % | m | % |
| 2 | RNG-Constant Mixing Length | 1,188 | -1,100 | -1.40 | -0.08 | -3.60 | -0.09 | -0.17 |
| 3 | k-epsilon | 726 | -2,400 | -3.05 | -0.11 | -4.80 | -0.71 | -1.38 |
| 4 | Prandtl Mixing Length | 816 | -200 | -0.25 | -0.09 | -3.91 | 0.43 | 0.83 |
| 5 | One-Equation-Constant Mixing Length | 1,190 | -700 | -0.89 | -0.09 | -4.02 | 0.42 | 0.80 |
| 6 | LES | 3,047 | -6,600 | -8.39 | -0.25 | -10.4 | -1.9 | -3.64 |
| 7 | Laminar | 3,386 | 5,200 | 6.61 | 0.09 | 3.60 | 0.81 | 1.57 |

**Table 2. Overtopping characteristics of three simulated avalanche events of different size for the current lake level and lake lowering scenarios (after Somos-Valenzuela et al., 2016).**

| Lake lowering | Boundary condition | Overtopping | Avalanche size | | |
|---|---|---|---|---|---|
| | | | Large | Medium | Small |
| Baseline (0 m lower) | Avalanche | Volume ($10^6\ m^3$) | 1.80 | 0.50 | 0.15 |
| | | Peak discharge ($m^3/s$) | 63,400 | 17,100 | 6,410 |
| | | Wave height (m) | 21.7 | 12.0 | 7.1 |
| | Mass-momentum | Volume ($10^6\ m^3$) | 1.64 | 0.15 | 0.014 |
| | | Peak discharge ($m^3/s$) | 54,600 | 6,000 | 592 |
| | | Wave height (m) | 15.9 | - | - |
| 15 m lower | Avalanche | Volume ($10^6\ m^3$) | 1.60 | 0.20 | 0.02 |
| | | Peak discharge ($m^3/s$) | 60,200 | 6,370 | 1,080 |
| | | Wave height (m) | 38.4 | 27.5 | 25.1 |
| | Mass-momentum | Volume ($10^6\ m^3$) | 0.83 | 0.034 | 0 |
| | | Peak discharge ($m^3/s$) | 25,700 | 1,510 | 0 |
| | | Wave height (m) | 32.0 | 25.4 | 0 |
| 30 m lower | Avalanche | Volume ($10^6\ m^3$) | 1.30 | 0.05 | 0 |
| | | Peak discharge ($m^3/s$) | 48,500 | 1,840 | 0 |
| | | Wave height (m) | 60.8 | 42.5 | 0 |
| | Mass-momentum | Volume ($10^6\ m^3$) | 0.45 | 0 | 0 |
| | | Peak discharge ($m^3/s$) | 15,100 | 0 | 0 |
| | | Wave height (m) | 46.1 | 0 | 0 |

**Table 3. Comparison of maximum wave heights for FLOW 3D and empirical calculations.**

| Avalanche size | Boundary condition | Max. wave height (m) | | Distance to peak (m) |
|---|---|---|---|---|
| | | Empirical | FLOW 3D | FLOW 3D |
| Large | Avalanche | 42 | 47.8 | 1080 |
| | Mass-Momentum | | 46.4 | 1039 |
| Medium | Avalanche | 21 | 30.1 | 318 |
| | Mass-Momentum | | NA | NA |
| Small | Avalanche | 9 | 19.6 | 108 |
| | Mass-Momentum | | NA | NA |

**Table 4. Characterization of scenario according to the volume of overtopping discharge. Scenarios labeled as "High discharge" had overtopping volumes > 20 x $10^3$ m³. Scenarios labeled as "Medium discharge" had overtopping volumes ≤ 20 x $10^3$ m³. Scenarios labeled as "No discharge" did not result in any overtopping.**

| Avalanche size | Boundary condition | Lake-lowering | | |
|---|---|---|---|---|
| | | 0 m | 15 m | 30 m |
| Large | Avalanche | High discharge | High discharge | High discharge |
| | Mass-momentum | High discharge | High discharge | High discharge |
| Medium | Avalanche | High discharge | High discharge | High discharge |
| | Mass-momentum | High discharge | Medium discharge | No discharge |
| Small | Avalanche | High discharge | Medium discharge | No discharge |
| | Mass-momentum | Medium discharge | No discharge | No discharge |

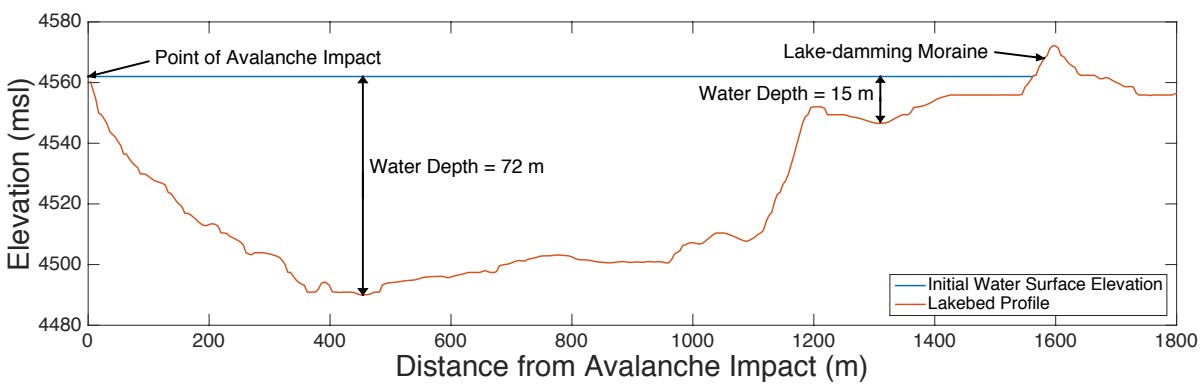

Figure 1: Location of Lake Palcacocha within the Cordillera Blanca, Peru.

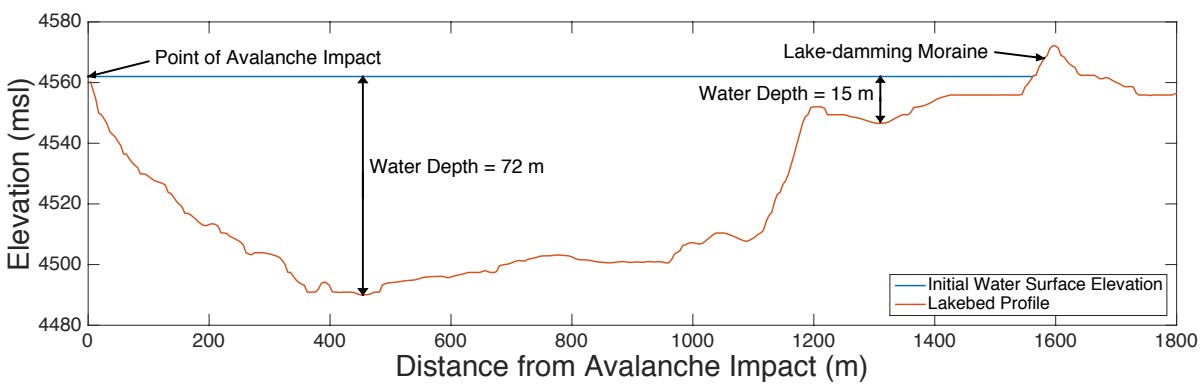

Figure 2. Longitudinal profile of Lake Palcacocha and its terminal moraine.

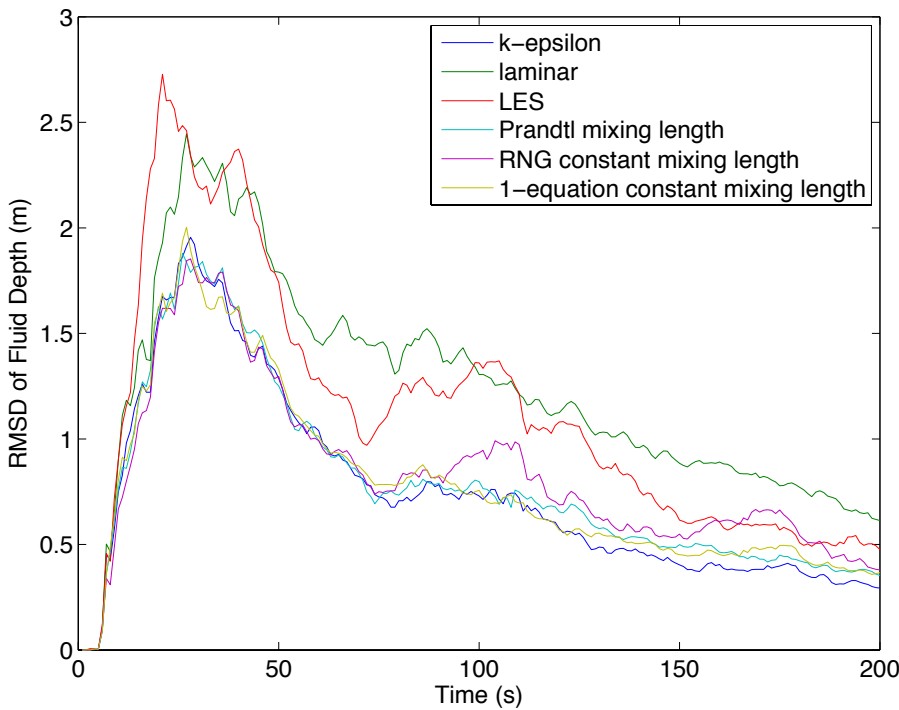

**Figure 3. Root-mean-square deviation (RMSD) of fluid depth from the baseline model results (RNG-dynamically computed mixing length) for each turbulence model as a function of time.**

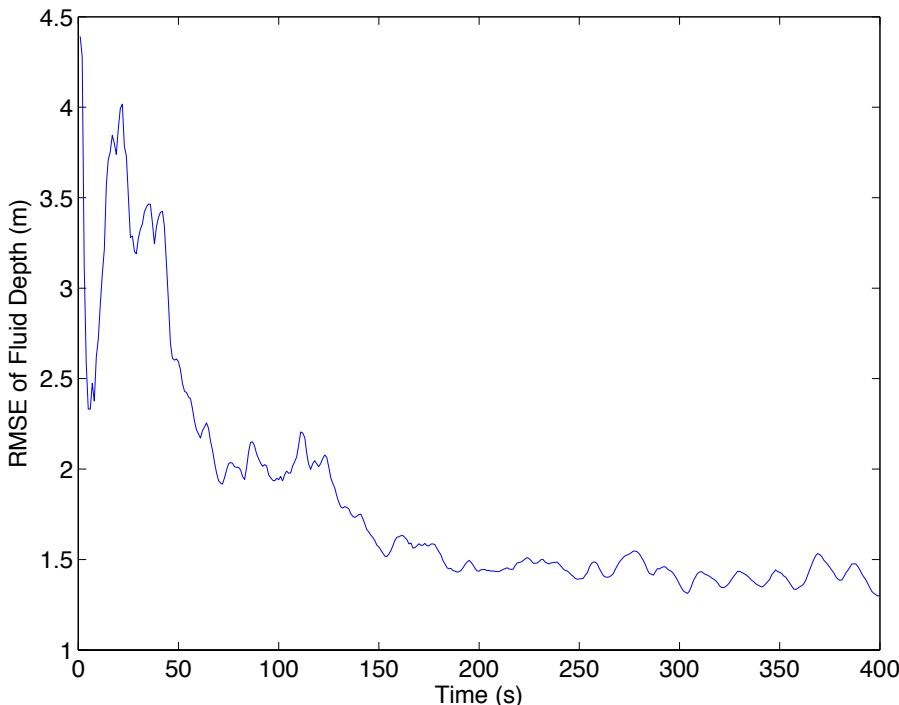

**Figure 4. RMSE of fluid depth for the coarse grid simulation as compared to the regular grid mesh using the baseline turbulence model.**

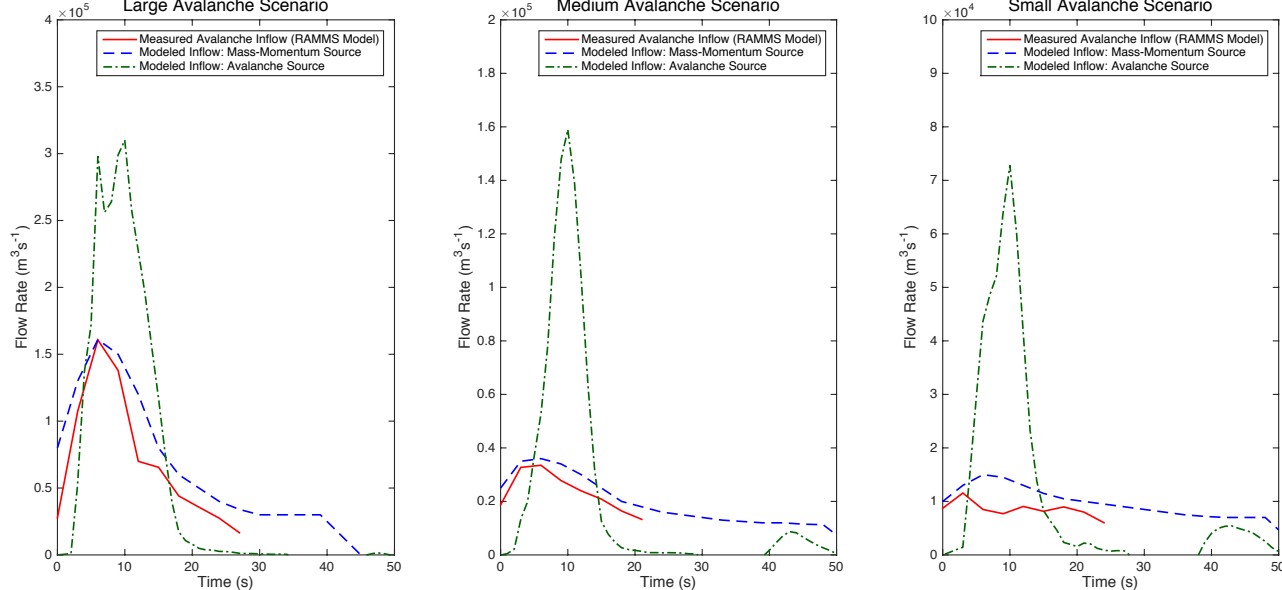

**Figure 5. Inflow hydrographs for the avalanche as it enters the lake for the avalanche source and mass-momentum source boundary conditions as compared to the hydrograph extracted from the RAMMS avalanche model.**

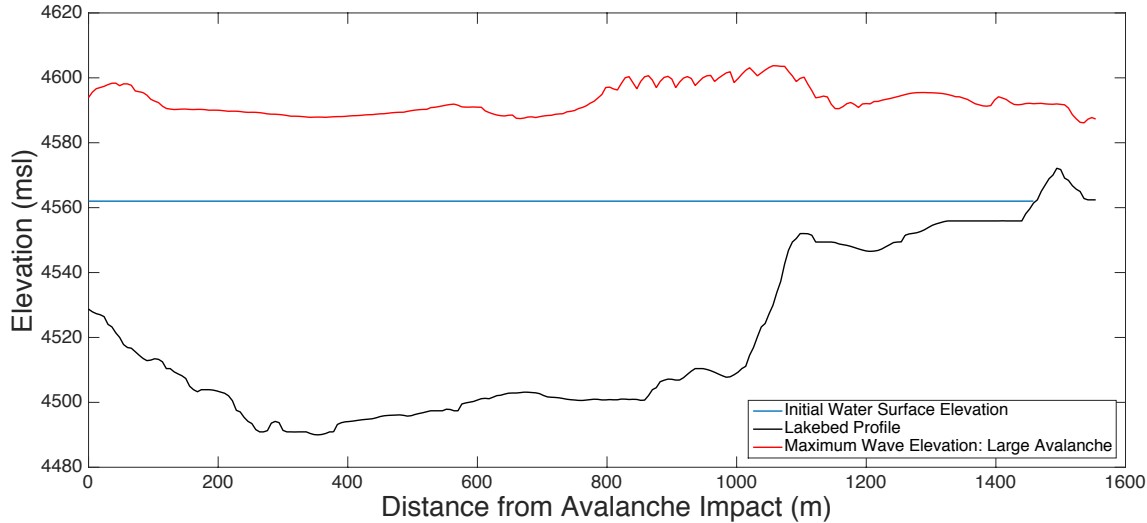

**Figure 6. Profile of the maximum wave height as a function of distance along the lake for the large avalanche boundary condition.**

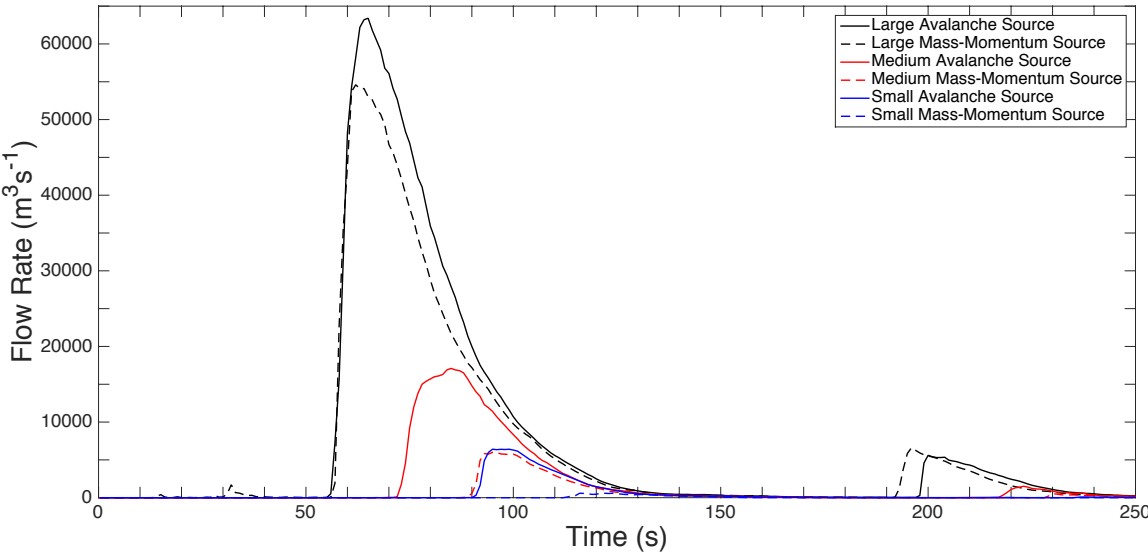

**Figure 7. Overtopping wave discharge hydrographs for the three avalanche events and two types of boundary conditions with the lake at the baseline level.**

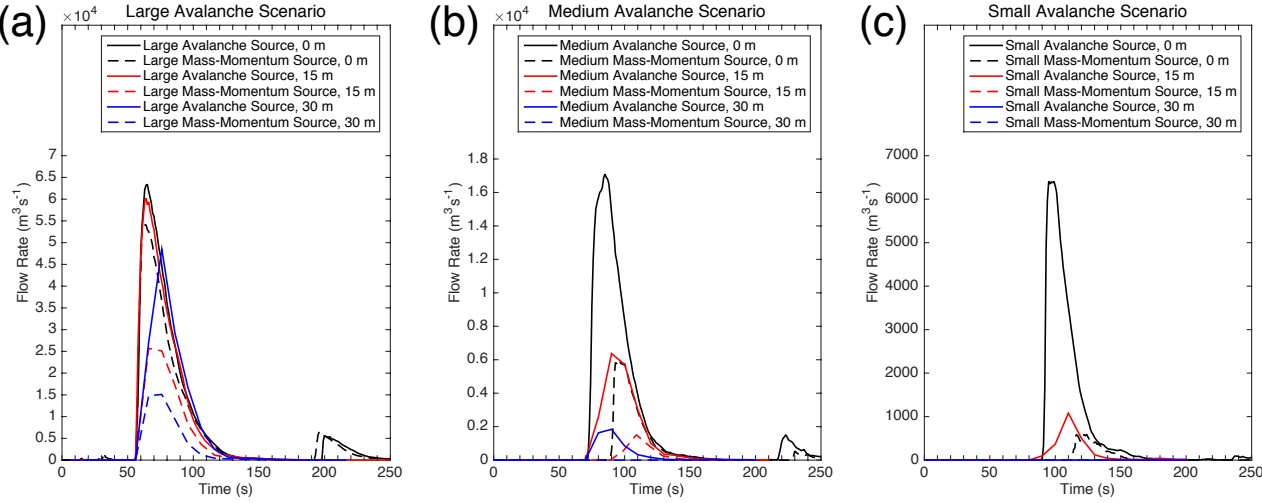

**Figure 8. Overtopping hydrographs for lake lowering scenarios for (a) large avalanche scenario, (b) medium avalanche scenario, and (c) small avalanche scenario.**