# Peer review of "Dynamics of avalanche-generated impulse waves: three-dimensional hydrodynamic simulations and sensitivity analysis"

_Natural Hazards and Earth System Sciences, 2017_

## Referee Comment (RC1) · A. Emmer (Referee) · 11 Jul 2017

Review to the manuscript 'Three-dimensional hydrodynamic lake simulations of avalanche-generated impulse wave dynamics for potential GLOF scenarios at Lake Palcacocha, Peru' submitted by Rachel E. Chisolm and Daene C. McKinney to NHESS

General comments:

Lake Palcacocha – an emblematic case in GLOF studies – has been attracting attention of scientists as well as local authorities and practitioners since catastrophical outburst in 1941 (e.g., Oppenheim, 1946), resulting in implementation of remedial works (open cut and two artificial dams) in 1970s (Zapata, 2002; Emmer et al., accepted). Since that time, the lake has increased its volume from ca 0.5 Mm$^3$ to about 17 Mm$^3$ nowadays (UGRH, 2016), due to the glacier retreat, making these mitigation measures insufficient. Recently, Emmer et al. (2016) did lake inventory of lakes of the Cordillera Blanca (882 lakes identified, classified and described by the set of qualitative and quantitative characteristics) and assessment of susceptibility to outburst floods of all large lakes (A>100,000 m$^2$; n=64), revealing that Lake Palcacocha is among those lakes ranked as "highly susceptible" to produce GLOF, moreover located upstrem regional capital Huaráz. This study on the level of mountain range identified fast slope movements into the lake as likely trigger of GLOF from Lake Palcacocha, but does not provide more detailed information on them – apparent research gap which needs to be addressed. From this point of view is research on potential slope movements entering the Lake Palcacochca undoubtedly desirable.

The authors of the presented manuscript apply known methods and software in geographically relatively new contex of Lake Palcacocha, focusing on ucertainities in modelling displacement wave dynamics (formation, propagation and dam overtopping). Comparison of results obtained from different methods / models is presented, however, with no validation against real GLOF event. This I found to be the major drawback of presented study – no real baseline is used, therefore models are compared to each others, but the overall value of obtained results and implications is, thus, somewhat uncertain.

Additionally, I'm convinced, that uncertainty of scenarios, methods and software used in entire process chain should be somehow balanced. Presented manuscript is, however, very much concerned about the uncertainities of methods and software used to simulate displacement wave dynamics, without considering factuality (uncertainty) of avalanche scenarios used. Why 0.5, 1.0 and 3.0 Mm$^3$ ?? Why not e.g., 0.2, 2.0 and 5.0 Mm$^3$ ?? Are there any field / remote sensing-based observations suggesting these volumes ?? This is major issue which needs to be addressed, otherwise all hazard mitigation implications and conclusions are rather speculative.

Moreover, Somos-Valenzuela et al. (2016) recently published modelling of entire process chain of outburst flood from Lake Palcacocha in HESS, considering the same ice avalanche and lake level lowering scenarios and using the same methods and software (Heller and Hager, 2010; FLOW3D, Flow Science, 2012). Releasing of this publication and submission of presented manuscript seems to me chronologically in reverse order, in other words, presented manuscript seems bit redundant now, when paper of Somos-Valenzuela et al. (2016) is published, despite the fact that presented manuscript provides "improved understanding of the dynamics of avalanche-generated waves". In conclusions, would presented results change the results of Somos-Valenzuela et al. (2016) in a significant way ?? Some of results presented are actually overlapping with Somos-Valenzuela et al. (2016), without mentioning it (the results presented in the first paragraph in section 3.4 are identical to results in the last paragraph in section 4.2.1 of Somos-Valenzuela et al. (2016)).

Novelty and additional value of presented manuscript compared to previous studies, therefore, need to be clearly shown, not only in terms of modelling (comparison of different approaches) but also in context of ongoing mitigation activities and research at Lake Palcacocha and realistic (observation-based) potential GLOF triggers.

Specific comments and technical notes (key ones in **bold**):

P01L13: consider using the term "displacement wave" in the manuscript

P02L11-16: see also recent lake inventory and GLOF susceptibility assessment for the Cordillera Blanca of Emmer et al. (2016)

P02L23: "... failure of lake-damming moraine."

P02L24: see the work of Shiva Pudasaini and r.avaflow project (http://www.avaflow.org)

P03L04: "... may overtop or breach moraine dams."

P03L28: replace "potential" by "future"

P03L35-36: this needs to be specified in more detail; this has already been done as a part of Somos-Valenzuela et al. (2016)

P04L07: replace "mostly destroyed" by "breached"

P04L09: replace "a smaller" by "basal"

P04L09: replace "back" by "upstream"; for people who don't know this area, **field-based figure for better imagination** on that would be nice

P04L12: lake growth (glacier retreat) 200 m retreat since 2009 ?? please check

P04L17: replace "terminal" by "damming"

P04L19-32: here, I miss two recent works focusing on Lake Palcacocha: Emmer et al. (2016) identifying Lake Palcacocha as highly susceptible and Klimeš et al. (2016) elaborating impact of potential landslides in moraines on the Lake Palcacocha (sorry, I contributed to both)

P04L01-P05L15: see also Westoby et al. (2014, 2015), Pudasaini & Hutter (2007); Mergili et al. (2016), r.avaflow project (r.avaflow.org) and others

P05L32-33: if it is true that the glacier retreated 200 m since 2009 (see P04L12), 2016 lake bathymetry should be used in order to obtain meaningful results

**P06L05: how do you know these are "likely avalanche sizes" ?? Why 0.5, 1.0 and 3.0 Mm$^3$ ?? Why not e.g., 0.2, 2.0 and 5.0 Mm$^3$ ?? Are there any field / remote sensing-based observations suggesting that ?? please explain and elaborate in detail**

P06L07: why this baseline ?? this explanation ("appropriate mixing length was unknown") is not clear to me

P07L17-L24: see also Westoby et al. (2014, 2015), Pudasaini & Hutter (2007); Mergili et al. (2016), r.avaflow project (r.avaflow.org) and others

P07L35-P08L02: this implication is not clear to me, please explain in more detail

P08L25: replace "terminal moraine" by "dam"

**P08L29: why not to use 2003 GLOF for the validation ?? yes, it was not ice avalanche-triggered GLOF (landslide in lateral moraine), but formation, propagation of displacement wave and dam overtopping occurred; please comment on that**

P09L18-L20: is that what is actually being done currently ?? if hazard mitigation implications are elaborated, more info on ongoing works should be provided

P13L08-L16: similar to last paragraph in section 4.2.1 of Somos-Valenzuela et al. (2016)

P14L01: replace "damming moraine" by "dam"

P14L18-P17L11: this is hard to follow; I strongly recommend to **structure discussion into the sub-sections reflecting individual issues discussed**

P14L31-L33: see also avaflow.org

P15L02-L07: comparing different models between each others with no comparison to reality seems slightly purposeless to me

**P16L19: from my point of view, the greatest uncertainity arises from not knowing realistic (field investigation-based) scenarios of potential ice avalanches entering Lake Palcacocha, making all hazard mitigation implications and conclusions rather speculative; please discuss that**

P17L06-L08: please add reference

P17L13-L14: improving the understanding of lake dynamics during GLOFs does not necessary require the use of 3D non-hydrostatic models only, this is just an option; please reformulate

P17L16-L17: this is very apparent

P17L22-L24: hazard level for Huaráz is not the subject of the manuscript, this conclusion is more based on work of Somos-Vlaenzuela et al. (2016); please rerformulate or avoid

Simulation vs. modelling – is there any difference ?? please explain and unify in manuscript

**Tab 2: if the wave height is defined as "height above the moraine crest as the wave overtops the damming moraine" (P13L10) it is not clear to me, why is increasing with decreasing lake level (volume, peak discharge) ?? please elaborate in more detail**

Based on above mentioned I suggest major revisions of the manuscript. I'll be happy to review revised version. Please, do not hesitate to contact me in case of questions (aemmer@seznam.cz).

Kind regards

Adam Emmer

References (style not unified, sorry):

Emmer A, Vilímek V, Zapata ML (2016, online first). Hazard mitigation of glacial lake outburst floods in the Cordillera Blanca (Peru): the effectiveness of remedial works. Journal of flood risk management. doi: 10.1111/jfr3.12241.

Emmer A, Klimeš J, Mergili M, Vilímek V, Cochachin A (2016). 882 lakes of the Cordillera Blanca: an inventory, classification and assessment of susceptibility to outburst flood. Catena 147: 269–279.

Flow Science. FLOW-3D Documentation: Release 10.1.0, Flow Science, Inc. Santa Fe, NM, 2012.

Heller, V., Hager, W. H. Impulse Product Parameter in Landslide Generated Impulse Waves. J. Waterw. Port CASCE, 136, 145-155, 2010.

Klimeš, J., Novotný, J., Novotná, I., Jordán de Urries, B., Vilímek, V., Emmer, A., Strozzi, T., Kusák, M., Cochachin Rapre, A., Hartvich, F., Frey, H. (2016): Landslides in moraines as triggers of glacial lake outburst floods: example from Palcacocha Lake (Cordillera Blanca, Peru). Landslides, 13(6): 1461-1477, doi: 10.1007/s10346-016-0724-4.

Mergili M, Fischer J-T, Krenn J, Pudasaini SP 2016. r.avaflow v1, an advanced open source computational framework for the propagation and interaction of two-phase mass flows, Geo-scientific Model Development 10: 553–569.

Oppenheim, V., 1946. Sobre las Lagunas de Huaraz. Boletin de la Sociedad Geologica del Peru 19, 68–80.

Pudasaini SP, Hutter K 2007. Avalanche Dynamics: Dynamics of rapid flows of dense granular avalanches. Springer.

Somos-Valenzuela, M. A., Chisolm, R. E., Rivas, D. S., Portocarrero, C., McKinney, D. C.: Modeling glacial lake outburst flood process chain: the case of Lake Palcacocha and Huaraz, Peru, Hydrol. Earth Syst. Sci., 20, 2519–2543, 2016. doi: 10.5194/hess-20-2519-2016

UGRH – Unidad de Glaciologia y Recursos Hídricos. Bathymetric survey of Lake Palcacocha. Autoridad Nacional de Agua (ANA) de Peru, Huaraz, Perú, 2016.

Westoby, M.J., Glasser, N.F., Brasington, J., Hambrey, M.J., Quincey, D.J., Reynolds, J.M., 2014. Modelling outburst floods from moraine-dammed glacial lakes. Earth Sci. Rev. 134, 137–159. http://dx.doi.org/10.1016/j.earscirev.2014.03.009.

Westoby, M.J., Brasington, J., Glasser, N.F., Hambrey, M.J., Reynolds, J.M., Hassan, M.A.A.M., Lowe, A., 2015. Numerical modelling of glacial lake outburst floods using physically based dam-breach models. Earth Surface Dynamics 3 (1), 171–199. http://dx.doi.org/10.5194/esurf-3-171-2015.

Zapata ML 2002. La dinamica glaciar en lagunas de la Cordillera Blanca, Acta Montana (ser. A Geodynamics) 19: 37–60.

---

## Referee Comment (RC2) · W. Schwanghart (Referee) · 12 Jul 2017

The manuscript has many overlaps with the HESS paper by Somos-Valenzuela et al. (2016). Yet, there is sufficient novelty and originality that warrant publication of this manuscript for which NHESS is a suitable outlet. This added value, however, should be addressed more rigorously before the manuscript is ready for publication.

The main strength of the paper lies in the sensitivity analysis of numerical models

of displacement wave generation, propagation, run-up and overtopping. A revised manuscript should accentuate this issue, both in the title and overall focus. So far, the paper emphasizes the GLOF hazard of Lake Palcacocha and conveys a case study rather than a more general treatment of the problem. I am sure that this would help to demarcate the paper against the paper by Somos-Valenzuela et al. (2016).

A stronger focus on sensitivity analysis would entail to shorten several of the general issues raised in the introduction but rather extend on the issues in section 1.2. A clearer distinction should also be made between the sensitivity assessment and the scenarios. So far, uncertainties that arise from the avalanche simulation process are mixed with the assumptions about the size of the avalanches or the scenarios, in general.

Sensitivity analysis is closely related to the analysis of uncertainties and how they propagate to measures relevant for decision makers. The crisp values listed in Table 2 and 3, and classification of scenarios into "safe" and "not safe" appear at odds with a probabilistic assessment but should rather incorporate uncertainty characterization which could either be quantitative (Table 2,3) or qualitative (Table 4) and derived from the sensitivity analysis.

I think that my concerns about the current manuscript can be adressed but need substantial rewriting or restructuring. Moreover, additional analysis may be required. I thus recommend major revisions.

Specific comments:

1, 11: risk in the risk literature is defined as average loss per year. I'd replace the term risk with probability.

2, 2: perhaps rephrase: "..., and Schwanghart et al. (2016) showed that more than 68% of Himalayan hydropower projects are located on potential GLOF tracks".

2, 10: It is unclear what the 40% refer to. 40% more lakes than before 1986?

2, 19: Can you briefly explain, why this is so? Is there something special about the

moraines in the Cordillera Blanca? Or is the statement more general, i.e., that moraine-dammed lakes are more susceptible than bedrock or ice-dammed lakes?

2, 29: any reference to back the statement that lake dynamics remain one of the most problematic processes?

3, 6: The abbreviation SWE (shallow water equations?) has not been introduced before.

3, 18: specify "this area"

4, 5: Please provide more quantitative information on the moraines here.

4, 33: I'd avoid the term "complex" here, in particular since you continue with "To complicate matters further...". Complex is not complicated.

6, 10f: I am not an expert in computational fluid dynamics, and have problems understanding this part. To avoid that readers get lost here, please try to use plain language to explain the issues with the turbulence model. Otherwise, this part is extremely technical when compared to the preceding part of the paper.

6, 18: Here scenarios are mentioned, but are later explained in 2.4. Consider rearranging the headings.

7, 1f: What is the source of the elevation and bathymetry data?

7, 10: I think this should be interpolation, not extrapolation.

7, 16: not easy to solve: avoid subjective statements

8, 20: provide reference to empirical methods

8, 27f: This point has been mentioned before. I'd delete this part.

10, 8: Avoid interpretation here

10, 6: It is a bit unsatisfactory that there is a baseline level of error due to the extrapolation of the initional conditions to a coarser grid. Any chances to overcome these issues? Otherwise, it is difficult to separate the sensitivity to initial conditions and sensitivity to grid size.

12, 27: I would avoid the term tsunami in this context. Rather call it a displacement wave.

14, 19f: The first paragraph should rather be placed in the introduction or removed.

16, 14: remove somewhat

Figure 1: Consider using hillshading to better visualize topography (see Fig. 1 in Somos-Valenzuela et al. 2016). In addition, adding glaciated areas would be helpful.

---

## Referee Comment (RC3) · Anonymous Referee #3 · 21 Jul 2017

General remark I appreciate that several research teams work in this area and different techniques could lead to better understanding of process related to natural hazards. I believe that such paper about modelling will be very useful, nevertheless if there were already published quantifications which might be used to verify the presented model, they should be used. I will understand if such verification will be already to large extension of submitted paper, but in this case it should be considered in the Discussion. I mean the: height of the wave overtopping the dam in 2003 (8 meters)

[Figure]

or estimation of landslide volumes on the inner parts of lateral moraine (see papers Vilímek et al., 2005 and Klimeš et al. 2016).

Specific comments: P1 L29: I suggest: "......a large number of these lakes pose a hazard or risk..." This is general sentence about GLOFs worldwide, not only about Palcacocha lake.

P2 L18: not only moraine-dammed lakes could be mentioned here, but also bedrock-dammed lakes (due to possible dam overtopping).

P2 L19-23: do not forget about rock-slides or rock-avalanches here,

P3 P3-4: this sentence should consider not only overtopping of moraines, but also breaching of moraines and overtopping of bedrock dams.

P4 L5: the steepness should be specified (inner and outer slopes). And be careful, because the current moraine holding the water behind (in lake frontal part), which you named "smaller moraine" on P4 L8 is rather different (in the steepness and not only) from the older one which was in function before 1941. This is clear for those who know the area personally, but not for all. I strongly suggest you to add a photo, on which you can explain the situation (like FIG. 1, perhaps better taken from the right lateral moraine). There exist another publication directly from the area of interest (Novotná and Klimeš, 2014), where you will find parts dedicated to steepness of slopes; for instance: "Investigations of mechanical and strength properties showed that the inner moraine slopes maintain temporal slope stability despite their steep dip (very often above 50°), which exceeds the dip of tested strength parameters. Their values are around 40° ".

P8 L23: I like the fact that you consider the lake bathymetry and surrounding topography, but it is at least necessary to add the citation or better to show a bathymetry map.

P8 L29: this is inaccurate. At least some data we have from one event from March

19th 2003.

P31 L31-32: Sure, the discrepancy in volume estimation is a problem, but why should be difference in moraine-dammed and bedrock-dammed lake in the sense of dam over-topping. Please explain. The only difference I see is in the potential for moraine erosion during the overflow compared to the bedrock. (To be precise the Lake 513 has a small moraine on the top of the bedrock-dam.).

P15 L37: Another sentence where the bathymetry map should be cited (or better included).

P 16 L18 This is OK for me, but there is probably one more issue which might be considered in the Discussion. Water will overtop the basal moraine first (you called them "small moraine") and soon it will reach the breakthrough (from 1941) in the former frontal moraine, which is rather narrow for the fluent continuation of the flood wave. (This is another reason why the photo should be included – perhaps taken from the right lateral moraine, that the former outburst will be clear visible).

P16 L31 Please consider, that the current dam of the Palcacocha Lake is rather wide in the foundation, compared to the typical narrow and high moraines from some other lakes which could be rather easily eroded during the overflow (or outburst).

P17 L19 Better to add the volumes for small and medium avalanches.
* * *
[Figure]

**Fig. 1.**

---

## Author Comment (AC1) · 9 Oct 2017

**Response to Reviewers nhess-2017-98**

**Three-dimensional hydrodynamic lake simulations of avalanche-generated impulse wave dynamics for potential GLOF scenarios at Lake Palcacocha, Peru**
**Rachel E. Chisolm and Daene C. McKinney**

**Reviewer 1**

General comments:

Lake Palcacocha – an emblematic case in GLOF studies – has been attracting attention of scientists as well as local authorities and practitioners since catastrophical outburst in 1941 (e.g., Oppenheim, 1946), resulting in implementation of remedial works (open cut and two artificial dams) in 1970s (Zapata, 2002; Emmer et al., accepted). Since that time, the lake has increased its volume from ca 0.5 Mm3 to about 17 Mm3 nowadays (UGRH, 2016), due to the glacier retreat, making these mitigation measures insufficient. Recently, Emmer et al. (2016) did lake inventory of lakes of the Cordillera Blanca (882 lakes identified, classified and described by the set of qualitative and quantitative characteristics) and assessment of susceptibility to outburst floods of all large lakes (A>100,000 m2; n=64), revealing that Lake Palcacocha is among those lakes ranked as "highly susceptible "to produce GLOF, moreover located upstream regional capital Huaráz. This study on the level of mountain range identified fast slope movements into the lake as likely trigger of GLOF from Lake Palcacocha, but does not provide more detailed information on them – apparent research gap which needs to be addressed. From this point of view is research on potential slope movements entering the Lake Palcacochca undoubtedly desirable.

The authors of the presented manuscript apply known methods and software in geographically relatively new context of Lake Palcacocha, focusing on uncertainties in modelling displacement wave dynamics (formation, propagation and dam overtopping). Comparison of results obtained from different methods / models is presented, however, with no validation against real GLOF event. This I found to be the major drawback of presented study – no real baseline is used, therefore models are compared to each other, but the overall value of obtained results and implications is, thus, somewhat uncertain.

Additionally, I'm convinced, that uncertainty of scenarios, methods and software used in entire process chain should be somehow balanced. Presented manuscript is, however, very much concerned about the uncertainties of methods and software used to simulate displacement wave dynamics, without considering factuality (uncertainty) of avalanche scenarios used. Why 0.5, 1.0 and 3.0 Mm3 ?? Why not e.g., 0.2, 2.0 and 5.0 Mm3 ?? Are there any field / remote sensing-based observations suggesting these volumes ?? This is major issue which needs to be addressed, otherwise all hazard mitigation implications and conclusions are rather speculative.

Moreover, Somos-Valenzuela et al. (2016) recently published modelling of entire process chain of outburst flood from Lake Palcacocha in HESS, considering the same ice avalanche and lake level lowering scenarios and using the same methods and software (Heller and Hager, 2010; FLOW3D, Flow Science, 2012). Releasing of this publication and submission of presented

manuscript seems to me chronologically in reverse order, in other words, presented manuscript seems bit redundant now, when paper of Somos-Valenzuela et al. (2016) is published, despite the fact that presented manuscript provides "improved understanding of the dynamics of avalanche-generated waves". In conclusions, would presented results change the results of Somos-Valenzuela et al. (2016) in a significant way ?? Some of results presented are actually overlapping with Somos-Valenzuela et al. (2016), without mentioning it (the results presented in the first paragraph in section 3.4 are identical to results in the last paragraph in section 4.2.1 of Somos-Valenzuela et al. (2016)).

Novelty and additional value of presented manuscript compared to previous studies, therefore, need to be clearly shown, not only in terms of modelling (comparison of different approaches) but also in context of ongoing mitigation activities and research at Lake Palcacocha and realistic (observation-based) potential GLOF triggers.

**Response to general comments from Reviewer 1:**
We would like to thank the reviewer for taking the time and effort to review this manuscript and provide thoughtful and constructive comments. We hope that this work is a contribution to the overall understanding of GLOF processes as they relate to hazard assessment. We agree with this reviewer that field data for model validation would be a valuable addition to this work, however, this was not possible due to the lack of field data that could be used for model validation (This is discussed further below in response to specific comments regarding this point). Therefore, it was necessary to find alternative ways to evaluate the uncertainties in the lake model. This paper attempts to study one of the numerous sources of uncertainty in GLOF process chain modeling by applying three-dimensional modeling methods to avalanche-generated waves. The idea behind this approach is to attempt to compensate for our lack of knowledge about the precise characteristics of avalanche-generated wave during actual GLOF events by investigating the model sensitivities and potential sources of uncertainty while relying less on calibration and more on physical representation of the wave dynamics by using a 3D non-hydrostatic model. By attempting to understand the potential range of uncertainty in the lake model, we aim to present the results of a lake model with a realistic representation of the limitations of these results and how they might be used in the context of hazard mitigation.

The authors agree with the reviewer and recognize that the avalanche is potentially the greatest source of uncertainty in the entire GLOF process chain, but the consideration of uncertainties due to the avalanche entering the lake is beyond the scope of this work. As there are no field-based measurements or remotely sensed data that could justify changing the avalanche sizes used in this work ($0.5$, $1$ and $3 \times 10^6$ m$^3$), we have decided to use values that are consistent with previously published studies for the Cordillera Blanca region. This work is an attempt to show a range of potential outcomes for an avalanche-triggered GLOF but does not address the uncertainties in the avalanche portion of the GLOF process chain. We do not currently possess the scientific knowledge to be able to precisely forecast or model avalanches, and investigating how avalanche dynamics are represented in models goes beyond the scope of this work. This paper addresses the uncertainties in the lake model itself and helps us gain a greater understanding of the parameters that are most likely to contribute to uncertainty in the lake model portion of a GLOF chain of events. We recognize that this is only one of many sources of uncertainty throughout the GLOF process chain, and evaluation of the relative contributions to

the overall uncertainty from the individual processes is beyond the scope of this work. One of the major conclusions of this work is that the specific characteristics of the avalanche and how they are represented in the model have the greatest impact on the overtopping discharges. At present, the methods for reducing the uncertainty in the avalanche model do not exist, and given the relative lack of field data on avalanche characteristics in the region, even quantification of uncertainty in the avalanche characteristics would be nearly impossible. Therefore, three sizes were selected for the avalanche simulations (presented in Somos-Valenzuela et al., 2016) that are consistent with the literature on avalanche-triggered GLOFs in the region. Perhaps equally notable, however, are the parameters that are not significant contributors of uncertainty in the lake model. For example, the lake model is not very sensitive to the turbulence model, therefore a modeler attempting similar work in the future can select a turbulence model with greater confidence that it will not significantly impact the results.

We agree with the reviewer that it would have been better to have release this paper before Somos-Valenzuela et al. (2016), as this work was a precursor and necessary input to the hazard mapping results presented in Somos-Valenzuela et al. (2016). However, due to time limitations, priority was placed on disseminating the results for the entire process chain model and hazard mapping (Somos-Valenzuela et al., 2016) because we believed that the results and conclusions could have implications for decision making and hazard mitigation strategies. The results for the avalanche source boundary condition were used as inputs for the downstream inundation model and hazard map in Somos-Valenzuela et al. (2016), but this paper also presents a second boundary condition method, the mass-momentum source. The reason for showing in this paper the lake model results that are also presented in Somos-Valenzuela et al. (2016) is to allow for comparison between the two boundary condition methods as well as to explain some of the nuances of how the modeling methods influence the lake dynamics that could not be explained in Somos-Valenzuela et al. (2016) due to space limitations. We believe that the method of representing the avalanche impact with the lake (boundary condition method) significantly affects the overtopping characteristics, and because we cannot say that one is more correct than another, we present both in order to give an idea of the range of uncertainty. As is discussed in this paper, we believe that the mass-momentum source boundary condition method may be underestimating the overtopping volumes while the avalanche source boundary condition method may be overestimating overtopping. Therefore, the actual overtopping discharges that could occur in a real GLOF event might be somewhat less than what is presented in Somos-Valenzuela et al. (2016), but we do not have enough information or confidence in the model results to be able to say this with certainty. While this does not substantially change the conclusions made in Somos-Valenzuela et al. (2016) (even small avalanches could present a significant hazard to the city of Huaraz), it does shed light on the level of uncertainty. The differences between the two boundary condition methods are more pronounced for smaller avalanches, and this range of uncertainty indicates that overtopping may or may not happen for the lake-lowering scenarios with small and medium avalanches. While Somos-Valenzuela et al. (2016) concluded that lake-lowering could reduce the overtopping volumes and inundation extents, the results in this paper present a more optimistic outlook for the impact of lake-lowering in that it may be possible to prevent overtopping for smaller avalanches. The suggestion for clarifying this in the conclusion is a good one and has been noted; we have modified the results and conclusions sections to include a reference to Somos-Valenzuela et al. (2016) and clarify how these results influence the conclusions that can be made.

This work is not intended to repeat what is presented in Somos-Valenzuela et al. (2016) but to present additional results and details about the lake model. This paper presents findings about the sensitivities and uncertainties of a three-dimensional lake model due to several types of inputs, including the type of boundary condition representing the avalanche impact with the lake and model parameters such as the turbulence model and grid size. We have attempted to present this work in a way that emphasizes the sensitivity analyses, uncertainty and aspects of the lake model that were not presented in Somos-Valenzuela et al. (2016). However, we understand that the novelty of this work and the distinction from Somos-Valenzuela et al. (2016) may not have come across clearly in the way this manuscript was written. We have made some changes to the manuscript to attempt to clarify this distinction. The authors believe that it is important to share the complete results and conclusions of the lake modeling efforts because the findings could have significant implications for GLOF process chain modeling and the proper selection of modeling methods for avalanche-generated waves.

Specific comments and technical notes (key ones in **bold**):
Responses to specific comments are given in blue

P01L13: consider using the term "displacement wave" in the manuscript
We agree with this suggestion, and it has been changed in the manuscript.

P02L11-16: see also recent lake inventory and GLOF susceptibility assessment for the Cordillera Blanca of Emmer et al. (2016)
We are familiar with this paper, and a reference has been added to the manuscript.

P02L23: "... failure of lake-damming moraine."
This change has been made, and the manuscript now reads, "the primary characteristics that signify a potentially hazardous glacial lake are the presence of overhanging ice and the likelihood of failure of the lake-damming moraine."

P02L24: see the work of Shiva Pudasaini and r.avaflow project (http://www.avaflow.org)
The authors have explored this model, and it appears that the model development has not yet reached a state to allow for 3D modeling of an avalanche-generated wave. When we discussed this with the model developers a couple of years ago, it was not yet ready for us to try, but it appears that the model is now available for download. The r.avaflow model looks promising in its capability to simulate multi-phase flows, and this could represent a significant advancement for the field of GLOF process chain modeling. However, this model seems to be focused on the interactions of the fluid and solid during impact and wave generation. The subsequent processes (wave propagation, run-up and overtopping) for avalanche-generated waves may be equally important to properly represent realistic overtopping volumes. Comparisons between 2D shallow water and 3D non-hydrostatic simulations indicate that the shallow water approximation is insufficient for representing the propagation and overtopping of this type of displacement wave (Heinrich, 1992; Zweifel et al., 2007; Somos-Valenzuela et al., 2016). The r.avaflow model uses depth-averaged equations for both the solid and fluid components of the flow. The use of a 2D

depth-averaged model for the propagation, run-up and overtopping phases may result in unrealistically optimistic overtopping volumes that could underrepresent the actual hazard levels. Nonetheless, we do believe that the r.avaflow model could be a great tool in GLOF process chain modeling, particularly for better understanding wave generation, and it is worth considering the use of this model in combination with 3D non-hydrostatic modeling of wave propagation in future studies of GLOF process chain modeling. We have been in communication with Shiva Pudasaini about this model, but unfortunately, the r.avaflow model was not ready for us to use when we did this work.

This model is an important contribution to the literature in the context of this work, and it was an oversight not to include references to it in the manuscript. We have added references to Pudasaini (2012) and Mergili et al. (2017) to the manuscript.

P03L04: "... may overtop or breach moraine dams."
We agree with this recommended change and have edited this to say: "These mass movement events can cause large waves that propagate across glacial lakes and may overtop or breach moraine dams"

P03L28: replace "potential" by "future"
We agree with this recommendation and have made this change in the manuscript.

P03L35-36: this needs to be specified in more detail; this has already been done as a part of Somos-Valenzuela et al. (2016)
We have added some text to this part of the manuscript to clarify the distinction between the two papers. The now states:
    "In this paper, Lake Palcacocha is used as a case study to investigate the impact of an avalanche event on the lake dynamics and the ensuing discharge hydrograph from the lake. This paper focuses exclusively on the lake dynamics for avalanche-generated waves and does not consider other parts of the GLOF process chain. The model results for the entire GLOF chain of events are presented in Somos-Valenzuela et al. (2016); some of the lake modeling results from Somos-Valenzuela et al. (2016) are presented here for the sake of comparison between methods and to allow for analysis of the model sensitivity to various input parameters."

P04L07: replace "mostly destroyed" by "breached"
We agree with this recommendation and have made this change in the manuscript.

P04L09: replace "a smaller" by "basal"
We have added "basal" to the description of the moraine that now retains the lake. This sentence now reads, "The original lake-damming terminal moraine was breached destroyed during the 1941 GLOF, and the lake is currently dammed by a smaller basal moraine that lies about 300 m upstream of the 1941 breach."

P04L09: replace "back" by "upstream"; for people who don't know this area, **field-based figure for better imagination** on that would be nice
We have changed "back" to "upstream", and we agree that this makes the description clearer.

P04L12: lake growth (glacier retreat) 200 m retreat since 2009 ?? please check
Rivas et al. (2105) documented the growth of Lake Palcacocha with Aster imagery, and the retreat of the lake between 2009 and 2012 can be seen in Figure 4 of Rivas et al. (2105). This reference has been added to the manuscript.

P04L17: replace "terminal" by "damming"
This has been changed to read, "The southern lateral moraine is prone to landslides into the lake, and a slide from this moraine in 2003 caused minor damage from a wave that overtopped a portion of the lake-damming moraine (Vilimek et al., 2005)."

P04L19-32: here, I miss two recent works focusing on Lake Palcacocha: Emmer et al. (2016) identifying Lake Palcacocha as highly susceptible and Klimeš et al. (2016) elaborating impact of potential landslides in moraines on the Lake Palcacocha (sorry, I contributed to both)
We agree that these are relevant sources, and these references have been added to the manuscript.

P04L01-P05L15: see also Westoby et al. (2014, 2015), Pudasaini & Hutter (2007); Mergili et al. (2016), r.avaflow project (r.avaflow.org) and others
References to r.avaflow (Pudasaini, 2012) and Mergili et al. (2017) have been added to the manuscript. Westoby et al., 2014 has already been included in the reference list, and the Westoby et al. (2015) and Pudasaini and Hutter (2007) references were not included because they focus on portions of the GLOF process chain (avalanche dynamics and moraine breaching) that are not considered in this paper. This comment also appears below for P7L17-24.

P05L32-33: if it is true that the glacier retreated 200 m since 2009 (see P04L12), 2016 lake bathymetry should be used in order to obtain meaningful results
The 2016 bathymetric survey was not undertaken until after the modeling work was completed, therefore it was not possible to include the bathymetric data from 2016 in the lake model.

**P06L05: how do you know these are "likely avalanche sizes" ?? Why 0.5, 1.0 and 3.0 Mm3 ?? Why not e.g., 0.2, 2.0 and 5.0 Mm3 ?? Are there any field / remote sensing-based observations suggesting that ?? please explain and elaborate in detail**
The avalanche scenarios used in this work are the same as the avalanche simulations published in Somos-Valenzuela et al. (2016). These avalanche sizes were selected based on other similar studies in the Cordillera Blanca region (e.g., Schneider et al., 2014) as well as considering a range of avalanche sizes that might be expected given the conditions of the glacier. While we recognize that other avalanche sizes are possible, these scenarios were selected to represent a range of potential outcomes and were not meant to be a comprehensive analysis of all possible avalanche sizes that could originate from the Palcaraju and Pucaranra glaciers. The justification for the three avalanche sizes used in this work is presented in more detail in Somos-Valenzuela et al. (2016).

P06L07: why this baseline ?? this explanation ("appropriate mixing length was unknown") is not clear to me
It was necessary to select a baseline model to which other turbulence model results could be compared in order to facilitate the analysis of the turbulence model sensitivity (the point of

comparison for all models must be consistent). The mixing length is a parameter in many turbulence models that must be specified by the user; it is a length scale used for the calculation of values within a turbulence model. Typically, a length scale for a turbulence model would be based on the flow conditions (e.g., depth of flow in a river), but because the conditions for avalanche-generated waves are so highly variable (both spatially and temporally), there is no length scale that would be an obvious choice for these simulations. Examples of some possible length scales include lake depth or wave height, but the correct length scale for a turbulence model depends highly on the flow conditions, and neither of these is likely an appropriate choice. The "dynamically-computed mixing length" is an option in some turbulence models that allows the model to determine the mixing length based on localized flow conditions (the mixing length varies throughout the simulation). For this reason, the RNG model with a dynamically-computed mixing length was selected as the point of comparison for other turbulence models because it eliminates the uncertainty of the mixing length parameter as a possible influence on the model results. It turns out that the model is not very sensitive to this parameter, but for consistency in comparisons between models, it was necessary to eliminate this variable in the baseline turbulence model.

P07L17-L24: see also Westoby et al. (2014, 2015), Pudasaini & Hutter (2007); Mergili et al. (2016), r.avaflow project (r.avaflow.org) and others
See response to the same comment on P4L1-P5L15

P07L35-P08L02: this implication is not clear to me, please explain in more detail
The momentum transfer from the avalanche to the lake is what generates the displacement wave. Thus, the wave characteristics depend both on the mass (equivalent to depth) and velocity of the avalanche as it enters the lake. If the avalanche source model can reasonably reproduce those characteristics of the avalanche in the flow entering the lake, then the representation of the wave generation in the model should reasonably approximate the wave generation characteristics that could occur during an actual GLOF event.

P08L25: replace "terminal moraine" by "dam"
The manuscript on P9L4 has been changed to read, "The primary parameters used to study the wave characteristics were the maximum height of the wave in the lake and the wave height as it overtopped the moraine dam."

**P08L29: why not to use 2003 GLOF for the validation ?? yes, it was not ice avalanche-triggered GLOF (landslide in lateral moraine), but formation, propagation of displacement wave and dam overtopping occurred; please comment on that**
In an ideal world, we would use data from past GLOF events such as the 2003 landslide into Lake Palcacocha from the lateral moraine to validate the model. However, we do not have enough data on the event itself to be able to do this. In order to use field data for model validation, we would need information on parameters such as wave heights, landslide characteristics, etc. It is true that we have some information on the 2003 landslide, but the only variable that is known with reasonable confidence is the landslide volume; we would also need information on the depth and/or velocity of the landslide to be able to estimate the momentum that would have been transferred to the lake to generate the overtopping wave. All we know for certain from that event is that the terminal moraine was overtopped (minimum run-up height of

~8 m) and the moraine structure was partially damaged, but we cannot back-calculate the characteristics of the overtopping wave to the level that would be needed for model validation. Additionally, the landslide entered the lake from the lateral moraine, so the direction of propagation would have been different from the avalanches considered here that would enter the lake along its primary axis (in the same direction as the long axis of the lake and perpendicular to the moraine dam). Thus, the wave from the 2003 landslide may have been reflected off the lateral moraines before reaching the damming moraine; this would be a source of energy dissipation that could reduce the overtopping discharge. A wave that enters the lake along the primary axis and is not reflected off of a lateral moraine could result in higher overtopping discharges.

P09L18-L20: is that what is actually being done currently ?? if hazard mitigation implications are elaborated, more info on ongoing works should be provided
Lake lowering is being considered by government officials, and based on personal conversations with those involved in the planning and decision-making, 30 m is the lowest lake lowering alternative that is being seriously considered, as further lake lowering (beyond 30 m) would likely not be feasible. More information on the downstream impacts of these scenarios is given in Somos-Valenzuela et al. (2016). However, as of yet, no decisions have been about concrete mitigation measures. One of the objectives of this work and the work presented in Somos-Valenzuela et al. (2016) is to provide information that can assist decision-makers in Huaraz with making more informed decisions about hazard mitigation alternatives. We are not, however, making definitive recommendations about what actions should be taken.

P13L08-L16: similar to last paragraph in section 4.2.1 of Somos-Valenzuela et al. (2016)
Because it was necessary to present some of the same results from Somos-Valenzuela et al. (2016) for the sake of comparison between the various input parameters and boundary conditions, the explanation of the results that we show is indeed similar. The parameters presented with the results in this section (Section 3.4) are key parameters for understanding the characteristics of the overtopping discharges, and in order for the reader to understand the results we present here, it was necessary to give some explanation of the parameters considered. They may be similar, and this is unavoidable, however, this paragraph does not directly repeat what was stated in section 4.2.1 of Somos-Valenzuela et al. (2016).

P14L01: replace "damming moraine" by "dam"
We believe that changing the term "damming moraine" to "dam" implies that the structure retaining the lake is entirely man-made, so we have decided to leave this as it is to ensure that readers understand that what holds the lake back is primarily natural moraine.

P14L18-P17L11: this is hard to follow; I strongly recommend to **structure discussion into the sub-sections reflecting individual issues discussed**
This is a good recommendation, and this section has been re-structured to include sub-headings for the different topics that are discussed.

P14L31-L33: see also avaflow.org
This is a relevant point, and a sentence has been added to the manuscript to discuss how multi-phase debris flow models such as r.avaflow relate to the work presented in this paper:

"Recent advances in two-phase flow models such as r.avaflow (Pudasaini, 2012; Mergili, 2017) can facilitate simulations of wave generation from avalanches entering a lake; however, the use of depth-averaged equations still limits the ability to use this type of model to simulate all of the phases of an avalanche-generated wave from wave generation to overtopping."

P15L02-L07: comparing different models between each others with no comparison to reality seems slightly purposeless to me

Unfortunately, there are insufficient field data for this type of avalanche-generated wave to allow for comparison of model results with actual GLOF events. Wave characteristics such as wave height and attenuation would be needed to do this comparison, but they do not exist. The types of data that we have from previous events such as the 2003 landslide at Lake Palcacocha generally consist of estimates of the total slide volume and the run-up heights of the waves on the terminal and/or lateral moraines. We can see in the 3D simulations of avalanche-generated waves that are presented in this paper that the wave heights and run-up heights are very different. The theoretical run-up height against a vertical wall is twice the wave height, but in natural environments, the actual run-up height is highly dependent on the field characteristics (eg., lakebed geometry and moraine slope). Additionally, for many GLOF events, even the run-up height is not known with certainty. In many cases, only the minimum run-up height is known because overtopping of the damming moraine was observed, so it was concluded that the run-up height must have been greater than the freeboard. In the case of the Lake 513 event, the wave height was estimated to be 25 m based on the 20 m freeboard of the rock dam and the 5 m of debris that were eroded by the overtopping wave; however, this is not an accurate estimate of the actual height of the avalanche-generated wave within the lake, and it is possible that the run-up height/overtopping wave height was greater than 25 m.

Because field data were not available, we had to look for alternatives to check the reasonableness of our results. The empirical model has been compared to measured data and laboratory experiments (a form of validation of the method), so it may reasonably be concluded that if the 3D model gives similar values for the wave characteristics, we can have more confidence in the 3D model. However, this comparison is only valid for the wave generation phase (maximum wave height), as the empirical model does not represent the wave propagation, run-up and overtopping phases well (if at all).

Because we have so little information about field characteristics of avalanche-generated waves, our best course of action to facilitate understanding of the wave dynamics is to use models that depend less on calibration (such as 2D SWE models) and more on better model representation of the physical processes (e.g., the 3D non-hydrostatic model that is used here). When model calibration and validation with field data is impossible, another necessary step is to investigate the model sensitivity to input parameters. This sensitivity analysis must be used in lieu of validation in order to better understand on the uncertainties of the model so that we do not represent more confidence in the model results than is justifiable given the uncertainties in the modeling process. Even though field data are not available to validate our model results, we believe that this should not prevent us from attempting to make progress in the methods for modeling avalanche-generated waves while still recognizing the limitations of our work.

**P16L19: from my point of view, the greatest uncertainty arises from not knowing realistic (field investigation-based) scenarios of potential ice avalanches entering Lake Palcacocha,**

**making all hazard mitigation implications and conclusions rather speculative; please discuss that**

This is true; we recognize that the avalanche is the portion of the GLOF process chain is the least understood and most likely the greatest source of uncertainty in GLOF modeling and hazard mapping. Although the uncertainty resulting from the avalanche portion of the chain of events must be considered in the decision-making process, investigating the uncertainty in the avalanche characteristics is beyond the scope of this work. This paper focuses exclusively on the lake dynamics of avalanche-generated waves and what are the potential sources of uncertainty in this portion of the GLOF process chain. In a way, the avalanche scenarios are attempting to capture some of that uncertainty, but it does not represent all of the uncertainty associated with the avalanche model. This work explores the uncertainties that arise when representing the impact of the avalanche with the lake and the wave generation, but this is necessarily limited by the avalanche characteristics that were available from the avalanche simulations. Until further advances are made in the field of avalanche simulations and we gain an improved understanding of avalanche dynamics for this type of event, it is impossible to incorporate all of the uncertainty of potential avalanches into analysis of the dynamics of avalanche-generated waves. All that we can do is assess the sensitivity of the lake model to different types of inputs to gain a qualitative understanding of how these uncertainties might impact the characteristics of avalanche-generated waves and the overtopping discharges.

Nonetheless, it is clear that steps to mitigate or reduce the hazard must be taken because even with the low end of the range of uncertainty in avalanche sizes, the resulting discharges could represent an unacceptable level of risk for the city of Huaraz. This is treated further with the hazard and inundation intensity maps for the various scenarios in Somos-Valenzuela et al. (2016).

We have modified the discussion section to more clearly convey this idea.

P17L06-L08: please add reference
A reference to Huggel et al. (2004) has been added to the manuscript.

P17L13-L14: improving the understanding of lake dynamics during GLOFs does not necessary require the use of 3D non-hydrostatic models only, this is just an option; please reformulate
We agree that the way this is worded is misleading, and this sentence has been changed in the manuscript to read: "Three-dimensional non-hydrostatic models can be a useful tool to simulate avalanche-generated waves and improve our understanding of lake dynamics during GLOF events."

While 3D non-hydrostatic models may not be required to improve our understanding of avalanche-generated waves, they are a very useful tool, especially given the lack of field observations. It is clear that 2D SWE models are insufficient for representing the dynamics of displacement waves, especially during the propagation, run-up and overtopping phases (Heinrich, 1992; Zweifel et al., 2007; Somos-Valenzuela et al., 2016). The hydrostatic approximation has a significant effect on the dissipation of energy, so hydrostatic models are likely to be severely underestimating overtopping discharges. Therefore, unless a calibration procedure is used similar to the one presented in Somos-Valenzuela et al. (2016) to compensate

for this dissipation of energy, a non-hydrostatic model is necessary to represent the dynamics of an avalanche-generated wave.

P17L16-L17: this is very apparent
Yes, it is obvious that larger avalanches pose the greatest threat, but the main point we were attempting to make is that even smaller avalanches could pose a threat that should not be ignored. The manuscript has been changed to read, "While large avalanches pose the greatest threat to the city of Huaraz, even smaller avalanches could generate significant overtopping discharges, resulting in substantial inundation in the city."

P17L22-L24: hazard level for Huaráz is not the subject of the manuscript, this conclusion is more based on work of Somos-Vlaenzuela et al. (2016); please rerformulate or avoid Simulation vs. modelling – is there any difference ?? please explain and unify in manuscript
That is correct; this paper does not address the downstream impacts of overtopping, as these are discussed in Somos-Valenzuela et al. (2016). The only conclusions that are made here refer to the overtopping of the lake-damming moraine. The point here is that overtopping could possibly be prevented (or reduced) for smaller avalanche scenarios through a lake-lowering mitigation project. It necessarily follows that if there is no overtopping, then potential hazards downstream could be prevented. It is not necessary to analyze downstream events to come to this conclusion. We may also conclude solely based on the results of the lake model that if overtopping discharges can be reduced by lake lowering, then the downstream hazard can also be reduced; exactly how much the downstream hazard can be reduced is not discussed in this paper and goes beyond the scope of this work (as it is presented in Somos-Valenzuela et al., 2016).

**Tab 2: if the wave height is defined as "height above the moraine crest as the wave overtops the damming moraine" (P13L10) it is not clear to me, why is increasing with decreasing lake level (volume, peak discharge) ?? please elaborate in more detail**
This is indeed a strange relationship, and upon first looking at it, it seems counterintuitive. However, we believe that this relationship of increasing wave heights above the moraine crest at the point of overtopping is physically correct. An explanation of this is given in the text (P14L11-17 of the manuscript) and has been modified to add additional clarity. The modified text is copied below.
> "The overtopping wave heights increased with lake lowering even though the total overtopping volumes and peak flow rates decreased. This may seem counterintuitive, but it can be explained by looking at how the lake dynamics may be expected to change with lake lowering. First, as the water surface level is lowered, the total volume stored in the lake increases, thus the momentum transferred to the lake from the avalanches per unit volume should be higher. The total volume in the lake decreases with lake lowering, so the additional momentum relative to the lake volume can produce taller waves. Secondly, as the point of avalanche impact is at a lower elevation relative to the avalanche release area with lake-lowering, there is more momentum in the avalanche fluid when it enters the lake. Although the increased overtopping wave heights for the lake lowering scenarios indicate that the waves may be larger when the lake is lowered, the amount of overtopping still decreases with lake lowering. This is most likely due to the lower initial water surface elevation; the lower free surface elevation results in a larger freeboard and means that more momentum is required for overtopping; although the momentum transfer per unit volume to the lake from the avalanche is greater, more of this momentum is lost during the run-up and overtopping, and less water is actually able to pass over the crest of the terminal moraine."

Based on above mentioned I suggest major revisions of the manuscript. I'll be happy to review revised version. Please, do not hesitate to contact me in case of questions (aemmer@seznam.cz).

Kind regards
Adam Emmer

References (style not unified, sorry):

Emmer A, Vilímek V, Zapata ML (2016, online first). Hazard mitigation of glacial lake outburst floods in the Cordillera Blanca (Peru): the effectiveness of remedial works. Journal of flood risk management. doi: 10.1111/jfr3.12241.

Emmer A, Klimeš J, Mergili M, Vilímek V, Cochachin A (2016). 882 lakes of the Cordillera Blanca: an inventory, classification and assessment of susceptibility to outburst flood. Catena 147: 269–279.

Flow Science. FLOW-3D Documentation: Release 10.1.0, Flow Science, Inc. Santa Fe, NM, 2012.

Heller, V., Hager, W. H. Impulse Product Parameter in Landslide Generated Impulse Waves. J. Waterw. Port CASCE, 136, 145-155, 2010.

Klimeš, J., Novotný, J., Novotná, I., Jordán de Urries, B., Vilímek, V., Emmer, A., Strozzi, T.,

Kusák, M., Cochachin Rapre, A., Hartvich, F., Frey, H. (2016): Landslides in moraines as triggers of glacial lake outburst floods: example from Palcacocha Lake (Cordillera Blanca, Peru). Landslides, 13(6): 1461-1477, doi: 10.1007/s10346-016-0724-4.

Mergili M, Fischer J-T, Krenn J, Pudasaini SP 2016. r.avaflow v1, an advanced open source computational framework for the propagation and interaction of two-phase mass flows, Geoscientific Model Development 10: 553–569.

Oppenheim, V., 1946. Sobre las Lagunas de Huaraz. Boletin de la Sociedad Geologica del Peru 19, 68–80.
Pudasaini SP, Hutter K 2007. Avalanche Dynamics: Dynamics of rapid flows of dense granular avalanches. Springer.

Somos-Valenzuela, M. A., Chisolm, R. E., Rivas, D. S., Portocarrero, C., McKinney, D. C.: Modeling glacial lake outburst flood process chain: the case of Lake Palcacocha and Huaraz, Peru, Hydrol. Earth Syst. Sci., 20, 2519–2543, 2016. doi: 10.5194/hess-20-2519-2016

UGRH – Unidad de Glaciologia y Recursos Hídricos. Bathymetric survey of Lake Palcacocha. Autoridad Nacional de Agua (ANA) de Peru, Huaraz, Perú, 2016.

Westoby, M.J., Glasser, N.F., Brasington, J., Hambrey, M.J., Quincey, D.J., Reynolds, J.M., 2014. Modelling outburst floods from moraine-dammed glacial lakes. Earth Sci. Rev. 134, 137–159. http://dx.doi.org/10.1016/j.earscirev.2014.03.009.

Westoby, M.J., Brasington, J., Glasser, N.F., Hambrey, M.J., Reynolds, J.M., Hassan, M.A.A.M., Lowe, A., 2015. Numerical modelling of glacial lake outburst floods using physically based dam-breach models. Earth Surface Dynamics 3 (1), 171–199. http://dx.doi.org/10.5194/esurf-3-171-2015.

Zapata ML 2002. La dinamica glaciar en lagunas de la Cordillera Blanca, Acta Montana (ser. A Geodynamics) 19: 37–60.

**Reviewer 2**

**General Comments**

The manuscript has many overlaps with the HESS paper by Somos-Valenzuela et al. (2016). Yet, there is sufficient novelty and originality that warrant publication of this manuscript for which NHESS is a suitable outlet. This added value, however, should be addressed more rigorously before the manuscript is ready for publication. The main strength of the paper lies in the sensitivity analysis of numerical models of displacement wave generation, propagation, run-up and overtopping. A revised manuscript should accentuate this issue, both in the title and overall focus. So far, the paper emphasizes the GLOF hazard of Lake Palcacocha and conveys a case study rather than a more general treatment of the problem. I am sure that this would help to demarcate the paper against the paper by Somos-Valenzuela et al. (2016).

A stronger focus on sensitivity analysis would entail to shorten several of the general issues raised in the introduction but rather extend on the issues in section 1.2. A clearer distinction should also be made between the sensitivity assessment and the scenarios. So far, uncertainties that arise from the avalanche simulation process are mixed with the assumptions about the size of the avalanches or the scenarios, in general. Sensitivity analysis is closely related to the analysis of uncertainties and how they propagate to measures relevant for decision makers. The crisp values listed in Table 2 and 3, and classification of scenarios into "safe" and "not safe" appear at odds with a probabilistic assessment but should rather incorporate uncertainty characterization which could either be quantitative (Table 2,3) or qualitative (Table 4) and derived from the sensitivity analysis.

I think that my concerns about the current manuscript can be addressed but need substantial rewriting or restructuring. Moreover, additional analysis may be required. I thus recommend major revisions.

**Response to General Comment from Reviewer 2:**
The authors would like to thank Reviewer 2 for the thoughtful feedback on this manuscript. The Reviewer's comment on the focus of this manuscript is correct. This paper is not meant to be a re-hashing of the lake model results from Somos-Valenzuela et al. (2016) but rather a complementary paper that focuses on the dynamics of avalanche-generated waves and the analysis of sensitivity to input parameters and boundary conditions. Very little work has been done in this area, especially with three-dimensional lake models, so we believe that the sensitivity analyses presented here can help future GLOF process chain modelers to better represent the dynamics of avalanche-generated waves and their overtopping characteristics. We recognize that the distinction between this paper and Somos-Valenzuela et al. (2016) may not have been clearly communicated in the manuscript, and are making changes to make this distinction more clear. We are considering the reviewer's suggestion to focus more on the sensitivity analysis in the revision of the manuscript, and we appreciate your specific suggestions on how to achieve this.

Regarding the definition of "safe" and "not safe" for the overtopping scenarios, we recognize that this terminology may not be accurately representing what we are trying to communicate. We are simply trying to clearly present which scenarios resulted in overtopping, but we do not intend this to replace probabilistic analysis of the scenarios and their relation to potential downstream hazards. The only thing we intend people to take away from this analysis is that if there is no overtopping, then that could prevent downstream inundation. Therefore, we are revising how this is presented in Table 4 to indicate the scenarios where overtopping did or did not occur. In addition, we are revising the accompanying explanatory text to more clearly communicate that these results to do not definitively indicate whether or not overtopping will occur but rather give a general idea of whether or not overtopping is likely. We do not attempt a probabilistic assessment of the hazard for each scenario nor a quantitative assessment of uncertainty, because it goes beyond the scope of this work, and we do not have sufficient information to do this. However, we would like to communicate more transparently what we are attempting to do and how this work relates to the larger problem of hazard assessment. Recognizing the limitations of our work is important to interpreting and applying the results, and we do not intend to imply more than what we have done in terms of hazard assessment and evaluation of mitigation alternatives. To fully analyze downstream impacts of mitigation alternatives and determine an optimum lake level requires consideration costs and benefits; further analysis beyond just the lake model is necessary, and this analysis is beyond the scope of this work. The overtopping discharges should not be considered in isolation when evaluating lake lowering and hazard mitigation alternatives. This paper does not attempt to address the decision-making process but only to provide information that might be useful inputs to that process.

In response to your comments, we are looking at ways to re-organize the paper and focus on the uncertainties and sensitivity analysis. While it is necessary to use some of the lake modeling results that are published in Somos-Valenzuela et al. (2016) for the sake of comparison with other methods and input parameters, we intend that this paper to be a separate work that shows what parameters in the lake model are most important to consider (where the model is most sensitive to inputs) and where the greatest sources of uncertainty in the lake model are likely to come from.

While there are many more parameters that we would like to be able to analyze, due to limitations in the time of the software license, we are not able to run any more simulations for further analysis.

**Specific comments:**
Responses to specific comments are given in blue.

1, 11: risk in the risk literature is defined as average loss per year. I'd replace the term risk with probability.
While there are many definitions of risk in the literature, you are correct that in the context of hazards, risk has a precise definition that includes hazard (combination of intensity and probability) and vulnerability. We believe that risk assessments should be the ultimate goal, but the work we do only deals with the hazard component. In this sentence, the intent is to convey a sense of the general threat to downstream communities from glacial lakes in the Cordillera Blanca, and we agree that while technically correct, the use of the word "risk" here may be misleading. Therefore, the word "risk" has been replaced with "threat" in the manuscript.

2, 2: perhaps rephrase: "..., and Schwanghart et al. (2016) showed that more than 68% of Himalayan hydropower projects are located on potential GLOF tracks".
This is a good suggestion that more clearly conveys the idea, and the manuscript has been changed accordingly.

2, 10: It is unclear what the 40% refer to. 40% more lakes than before 1986?
We agree that this wording is unclear and appreciate you pointing it out to us. The manuscript has been changed to read: "Cook et al. (2016) studied glaciers in the Bolivian Andes where glaciers have receded about 40% between 1986 and 2014, resulting in an increasing number of proglacial lakes."

2, 19: Can you briefly explain, why this is so? Is there something special about the moraines in the Cordillera Blanca? Or is the statement more general, i.e., that morainedammed lakes are more susceptible than bedrock or ice-dammed lakes?
In general, moraine-dammed lakes are more susceptible because failure of the moraine dam is an additional factor that could cause a GLOF (a potential GLOF trigger that is not present with rock-dammed lakes). Moraine dams are not exclusive to the Cordillera Blanca, but this region has a high number of lakes that are moraine-dammed, so the potential for moraine failure is a factor that should be considered as part of GLOF hazard assessment if a lake is moraine-dammed. Emmer and Cochachin (2013) further discuss the mechanisms of moraine-dam failures and how this varies from region to region. To clarify this point, the manuscript has been changed to:
   "Moraine-dammed lakes such as those present in the Cordillera Blanca are particularly susceptible to outburst flooding due to the potential for moraine failure that could cause higher GLOF discharges than would occur with just overtopping (Emmer and Cochachin, 2013); however, both moraine-dammed and bedrock-dammed lakes can produce potentially catastrophic GLOFs due to overtopping if there is insufficient freeboard."

2, 29: any reference to back the statement that lake dynamics remain one of the most problematic processes?

In general, lake dynamics for avalanche-generated waves have not been studied in detail, and we have very little in-situ data (almost none) from GLOF events. Several sources say that 2D SWE models don't represent the dynamics of this type of wave well (Heinrich, 1992; Zweifel et al., 2007; Worni et al., 2014; Somos-Valenzuela et al., 2016), but 3D models have not been used in this context before. Most process chain modeling studies use 2D SWE lake models (e.g., Schnieder et al., 2014; Worni et al., 2013). Because the hydrostatic approximation has a significant effect on the dissipation of energy, GLOF process chain simulations that use 2D SWE models are likely to be severely underestimating overtopping discharges. This paper attempts to address this research gap by investigating the use of a 3D non-hydrostatic lake model.

To clarify this point, the text in the manuscript has been changed to read, "however, the lake dynamics remain one of the most difficult processes to simulate correctly due to the need for a non-hydrostatic model to correctly represent the wave dynamics (Heinrich, 1992; Zweifel et al., 2007; Worni et al., 2014) and the lack of field data for model calibration or validation."

3, 6: The abbreviation SWE (shallow water equations?) has not been introduced before.
Thank you for pointing this out. It has been corrected in the manuscript.

3, 18: specify "this area"
This is indeed a vague reference, and the manuscript has been changed to read, "Huaraz is the most populous city in the Cordillera Blanca region"

4, 5: Please provide more quantitative information on the moraines here.
Information on the moraine slopes has been added to the manuscript as well as information on the width, height and width-to-height ratio of the lake-damming moraine. The edited text now reads, "The lake is surrounded on three sides by glacial moraines, and the lateral moraines are very tall with slopes up to 80º (Klimes et al., 2016). The southern lateral moraine is prone to landslides into the lake, and a slide from this moraine in 2003 caused minor damage from a wave that overtopped a portion of the lake-damming moraine (Vilimek et al., 2005). The original lake-damming terminal moraine was breached destroyed during the 1941 GLOF, and the lake is currently dammed by a smaller basal moraine that lies about 300 m upstream of the 1941 breach. This smaller moraine that currently holds back the lake is approximately 66 m deep, 985 m wide and has a width-to-height ratio of 14.9; this morphology indicates that the lake-damming moraine is very stable (Rivas et al., 2015)."

4, 33: I'd avoid the term "complex" here, in particular since you continue with "To complicate matters further...". Complex is not complicated.
It is a true statement that slide-generated waves have complex dynamics, and references from the literature have been added to the manuscript to support this statement (Worni et al., 2014; Fritz et al., 2004). The edited text in the manuscript now reads, "The dynamics of avalanche or landslide-generated waves are very complex (Fritz et al., 2004; Worni et al., 2014). In addition, it is very difficult to obtain field measurements of these waves to better understand their dynamics."

6, 10f: I am not an expert in computational fluid dynamics, and have problems understanding this part. To avoid that readers get lost here, please try to use plain language to explain the issues with the turbulence model. Otherwise, this part is extremely technical when compared to the preceding part of the paper.

We appreciate this feedback, especially from those who are less familiar with CFD because we would like for this to clear to all readers. We especially believe that it is important to communicate this clearly to non-CFD specialists because many involved in GLOF process chain modeling come from backgrounds other than engineering and computational fluid dynamics, and we hope that this paper can assist them in selecting appropriate modeling methods and efficiently concentrating their efforts to get more accurate estimates of overtopping discharges. While we don't want to lose any of the technical rigor of the turbulence model sensitivity analysis in how this is communicated, we want this to be understandable to both the CFD experts and scientists in other fields. The main point we want to convey is that we evaluated a number of turbulence models and determined that the uncertainty resulting from the turbulence model is small when compared to the uncertainty resulting from other aspects of the model. This portion of the manuscript is being revised to address this concern.

6, 18: Here scenarios are mentioned, but are later explained in 2.4. Consider rearranging the headings.

We believe that changing the order of the sub-sections here would not present things in a logical order (the scenarios really are the last thing considered and we use the work presented in prior sub-sections as a basis for the scenario analysis). To resolve the problem of referring to scenarios before they are mentioned, we have removed the reference to the scenario here and simply referred to the characteristics of the inputs used for the turbulence model sensitivity simulations.

7, 1f: What is the source of the elevation and bathymetry data?

The lake bathymetry used in this model comes from a 2009 bathymetric survey done by the Peruvian Glaciology and Water Resources Unit (called UGRH from the name of the unit in Spanish). This is stated at the beginning of Section 2 (P5L32-33). However, we neglected to state the source of the topographic data. The topography of the surrounding terrain comes from Horizons (2013). This reference has been added to the manuscript. We apologize for this oversight.

7, 10: I think this should be interpolation, not extrapolation.

This is not interpolation. We are simply assigning the same values to the 4 smaller cells that fall within each cell in the coarser grid. To clarify this, we have modified the text to read, "For the coarse grid simulations, the same value for water depth at each time step in the coarse grid was assigned to the four smaller cells that fall within each cell in the coarser grid to allow for direct comparison between the results of the coarse grid simulation and the regular mesh."

7, 16: not easy to solve: avoid subjective statements

This text has been modified to read, "The problem of reproducing an avalanche-generated impulse wave in a hydrodynamic model of a glacial lake presents a challenge because of the complicated dynamics of mixing and dissipation of energy that occur at the point of impact; these processes are difficult to represent correctly in the model."

8, 20: provide reference to empirical methods
A reference to the empirical method that is used in this work (Heller and Hager, 2010) has been added here.

8, 27f: This point has been mentioned before. I'd delete this part.
This portion of the text has been edited to remove redundancies.

10, 8: Avoid interpretation here
You are correct that interpretation is inappropriate here in the results section. This sentence has been changed in the manuscript to read, "The laminar model shows high deviations from the baseline model because it does not account for turbulence and should be the least dissipative of all the models."

10, 6: It is a bit unsatisfactory that there is a baseline level of error due to the extrapolation of the initional conditions to a coarser grid. Any chances to overcome these issues? Otherwise, it is difficult to separate the sensitivity to initial conditions and sensitivity to grid size.
This is simply because the bathymetry and thus initial still water depths cannot be represented as well with the coarser grid (the resolution of the bathymetry must be the same as the resolution of the model grid, so we lose some of the precision of the bathymetric representation that the finer grid has). We do not think there is any way to overcome this, but we have added a sentence to the manuscript to clarify this point. The modified text now reads: "However, there was a baseline level of error that comes simply from extrapolating the initial conditions to a coarser grid because the bathymetry and initial fluid depths are better represented in the fine grid model; this baseline error was unavoidable because the resolution of the bathymetry must be the same as the resolution of the model grid, so we lose some of the precision of the bathymetric representation in the model with the coarse grid."

12, 27: I would avoid the term tsunami in this context. Rather call it a displacement wave.
This change has been made in the manuscript, and it now reads, "The impact of the avalanche with the lake generates a large displacement wave."

14, 19f: The first paragraph should rather be placed in the introduction or removed.
This is a good suggestion, and this change has been incorporated into the re-organization of the manuscript.

16, 14: remove somewhat
This change has been made in the manuscript.

Figure 1: Consider using hillshading to better visualize topography (see Fig. 1 in Somos-Valenzuela et al. 2016). In addition, adding glaciated areas would be helpful.
We have considered this suggestion, and we decided that the current figure is adequate for the purposes of this paper.

**Reviewer 3**

**General Comment:**

General remark I appreciate that several research teams work in this area and different techniques could lead to better understanding of process related to natural hazards. I believe that such paper about modelling will be very useful, nevertheless if there were already published quantifications which might be used to verify the presented model, they should be used. I will understand if such verification will be already to large extension of submitted paper, but in this case it should be considered in the Discussion. I mean the: height of the wave overtopping the dam in 2003 (8 meters) or estimation of landslide volumes on the inner parts of lateral moraine (see papers Vilímek et al., 2005 and Klimeš et al. 2016).

**Response to General Comment from Reviewer 3:**
We would like to thank Reviewer 3 for the helpful comments on this manuscript. We agree with the point you make that validation of this model with field observations would be very useful. Unfortunately, the data we do have from previous GLOF events are insufficient for validation of this type of model. The 2003 landslide into Lake Palcacocha that you mention does have some information. In particular, we have a reasonable approximation of the landslide volume and the minimum run-up height over the lake-damming moraine complex. The estimate of the wave height in Klimes (2005) is based on the fact that the lake-damming moraine that had a freeboard of 8 m was overtopped. However, this would, in fact, be the minimum run-up or overtopping height and is different from the height of the wave within the lake. The actual height of the overtopping wave above the moraine crest is unknown. We have also found that the lake model is very sensitive to the characteristics of the mass movement into the lake (landslide or avalanche depth and velocity), and these parameters are not known precisely for the 2003 landslide at Lake Palcacocha. In order to do a meaningful comparison of this event with the results from the simulations of the lake model, we would need more information on the characteristics of the landslide other than just volume. This is something we would like to have been able to do, but unfortunately it was not possible with the available data.

However, your point about model validation is a good one, and we expect that many readers would have the same question that you present. We have added text to the discussion to address this issue more clearly.

**Specific comments:**
Responses to specific comments are given in blue.

P1 L29: I suggest: ": : :: : :a large number of these lakes pose a hazard or risk: : :" This is general sentence about GLOFs worldwide, not only about Palcacocha lake.
This change has been made to the manuscript, and the sentence now reads, "Glacier retreat worldwide has resulted in the emergence and growth of glacial lakes that have replaced ice in the tongue area of many glaciers, and a large number of these lakes pose a hazard or risk of glacial lake outburst floods (GLOFs)."

P2 L18: not only moraine-dammed lakes could be mentioned here, but also bedrock-dammed lakes (due to possible dam overtopping).

In general, moraine-dammed lakes are more susceptible because failure of the moraine dam is an additional factor that could cause a GLOF (a potential GLOF trigger that is not present with rock-dammed lakes), and this is all this sentence was meant to convey. However, you are correct that there is still a significant potential for GLOFs due to overtopping with bedrock-dammed lakes. The manuscript has been changed to clarify this and now reads,

"Moraine-dammed lakes such as those present in the Cordillera Blanca are particularly susceptible to outburst flooding due to the potential for moraine failure that could cause higher GLOF discharges than would occur with just overtopping (Emmer and Cochachin, 2013); however, both moraine-dammed and bedrock-dammed lakes can produce potentially catastrophic GLOFs due to overtopping if there is insufficient freeboard."

P2 L19-23: do not forget about rock-slides or rock-avalanches here,

You are correct that the potential for rock-slides or rock-avalanches are other factors that influence the potential for a lake to produce a GLOF. These factors have been added to the manuscript that now reads, "According to studies that have established basic methods for evaluating potential glacial lake hazards (e.g., Haeberli et al., 1989; Huggel et al., 2004; Wang et al., 2011; Emmer and Vilímek, 2014; Rounce et al., 2016), the primary characteristics that signify a potentially hazardous glacial lake are the presence of overhanging ice and the likelihood of failure of the lake-damming moraine; secondary characteristics that in indicate potentially dangerous lakes include the potential for landslides, rock-slides or rock-ice avalanches."

P3 P3-4: this sentence should consider not only overtopping of moraines, but also breaching of moraines and overtopping of bedrock dams.

The manuscript has been changed to reflect this suggestion, and this sentence now states, "These mass movement events can cause large waves that propagate across glacial lakes and may overtop or breach moraine dams or cause overtopping of bedrock dams."

P4 L5: the steepness should be specified (inner and outer slopes). And be careful, because the current moraine holding the water behind (in lake frontal part), which you named "smaller moraine" on P4 L8 is rather different (in the steepness and not only) from the older one which was in function before 1941. This is clear for those who know the area personally, but not for all. I strongly suggest you to add a photo, on which you can explain the situation (like FIG. 1, perhaps better taken from the right lateral moraine). There exist another publication directly from the area of interest (Novotny and Klimeš, 2014), where you will find parts dedicated to steepness of slopes; for instance: "Investigations of mechanical and strength properties showed that the inner moraine slopes maintain temporal slope stability despite their steep dip (very often above 50), which exceeds the dip of tested strength parameters. Their values are around 40 ".

Information on the moraine slopes has been added here. The type of moraine that currently dams the lake has been clarified in the manuscript (now referred to as a smaller basal moraine to distinguish it from the larger terminal moraine that was breached in 1941). The description of the moraines in the revised text is copied below:

"The lake is surrounded on three sides by glacial moraines, and the lateral moraines are very tall with slopes up to 80º (Klimes et al., 2016). The southern lateral moraine is prone to landslides into the

lake, and a slide from this moraine in 2003 caused minor damage from a wave that overtopped a portion of the lake-damming moraine (Vilímek et al., 2005). The original lake-damming terminal moraine was breached destroyed during the 1941 GLOF, and the lake is currently dammed by a smaller basal moraine that lies about 300 m upstream of the 1941 breach. This smaller moraine that currently holds back the lake is approximately 66 m deep, 985 m wide and has a width-to-height ratio of 14.9; this morphology indicates that the lake-damming moraine is very stable (Rivas et al., 2015)."

P8 L23: I like the fact that you consider the lake bathymetry and surrounding topography, but it is at least necessary to add the citation or better to show a bathymetry map.
The source of the bathymetry data (UGRH, 2009) is given in Section 1.1 (P4L11-12) and again in Section 2 (P5L32-33). The lake bathymetry used in this model comes from a 2009 bathymetric survey done by the Peruvian Glaciology and Water Resources Unit (called UGRH from the name of the unit in Spanish). In lieu of showing a bathymetric map, we have decided to show the profile of the lake (Figure 2), as we believe this better communicates the topography of the lakebed. This is stated at the beginning of Section 2 (P5L32-33). However, we neglected to state the source of the topographic data. The topography of the surrounding terrain comes from Horizons (2013). We apologize for this oversight., and a reference has been added with the source of the topographic data used in this work.

P8 L29: this is inaccurate. At least some data we have from one event from March 19th 2003.
You are correct that we do have some data from the 2003 landslide. However, the data we have from this event are insufficient for model validation. The types of data that we have from previous events such as the 2003 landslide at Lake Palcacocha generally consist of estimates of the total slide volume and the run-up heights of the waves on the terminal and/or lateral moraines. We can see in the 3D simulations of avalanche-generated waves that are presented in this paper that the wave heights and run-up heights are very different. The theoretical run-up height against a vertical wall is twice the wave height, but in natural environments, the actual run-up height is highly dependent on the field characteristics (eg., lakebed geometry and moraine slope). Additionally, for many GLOF events, even the run-up height is not known with certainty. In many cases, only the minimum run-up height is known because overtopping of the damming moraine was observed, so it was concluded that the run-up height must have been greater than the freeboard.

In response to this comment and similar ones from other reviewers, we have added a few sentences to the manuscript to clarify what data are available from the 2003 landslide and why we cannot use that event to validate the model.

P31 L31-32: Sure, the discrepancy in volume estimation is a problem, but why should be difference in moraine-dammed and bedrock-dammed lake in the sense of dam overtopping. Please explain. The only difference I see is in the potential for moraine erosion during the overflow compared to the bedrock. (To be precise the Lake 513 has a small moraine on the top of the bedrock-dam.).
The authors could not find the text referred to by the reviewer. There is no page 31 in the manuscript.

P15 L37: Another sentence where the bathymetry map should be cited (or better included).
This reference has been added to the manuscript.

P 16 L18 This is OK for me, but there is probably one more issue which might be considered in the Discussion. Water will overtop the basal moraine first (you called them "small moraine") and soon it will reach the breakthrough (from 1941) in the former frontal moraine, which is rather narrow for the fluent continuation of the flood wave. (This is another reason why the photo should be included – perhaps taken from the right lateral moraine, that the former outburst will be clear visible).
The potential erosion of the moraine was extensively considered in Somos-Valenauela et al (2016) and found to be negligible, and that analysis is not repeated here.

P16 L31 Please consider, that the current dam of the Palcacocha Lake is rather wide in the foundation, compared to the typical narrow and high moraines from some other lakes which could be rather easily eroded during the overflow (or outburst).
Consideration of the potential erosion of the damming-moraine is beyond the scope of this work. The potential erosion of the damming-moraine is discussed in Rivas et al. (2015) and Somos-Valenzuela et al. (2016). Somos-Valenzuela et al. (2016) determined that a breach of the damming-moraine due to erosion from the overtopping wave is very unlikely. This is for the reason that you mention in your comment: the high width-height ratio of the lake-damming moraine and the low slope of the distal face of the moraine (making it more stable).

P17 L19 Better to add the volumes for small and medium avalanches.
The avalanche volumes have been added to the manuscript.

---

## Referee Report (RR1)

Review to the revised version of the manuscript 'Three-dimensional hydrodynamic lake simulations of avalanche-generated impulse wave dynamics for potential GLOF scenarios at Lake Palcacocha, Peru' (now named 'Dynamics of avalanche-generated impulse waves: three-dimensional hydrodynamics simulations and sensitivity analysis') submitted by Rachel E. Chisolm and Daene C. McKinney to NHESS.

I'm glad to report that the authors have considered and replied to all issues arose by the reviewers and have edited their manuscript accordingly, leading to the substantial improvement. Most importantly, the authors have stressed rationale and the novelty compared to the paper of Somos-Valenzuela et al. (2016), have structured and extended the discussion section and conclusions and have explained unobvious statements in more detail. The authors have also reflected most of the minor comments and suggestions. I only have few minor comments and suggestions (see below).

Minor specific comments and technical notes:

P4L29: technically, the mitigation measure applied at the dam of Lake Palcacocha is not a tunnel, but the combination of open cut and artificial dam

P6L4: Mergili et al. (2017) is not in the list of references; r.avaflow has already been sucessfully employed to simulate GLOF process chain in the Cordillera Blanca (2012 event in Santa Cruz; see also http://onlinelibrary.wiley.com/doi/10.1002/esp.4318/full)

P16L20: 'as the water surface level is lowered, the total volume stored in the lake increases' - please check

P21L19-21: some passages are repeating, please check

P21L31: cannot be quantified using given approach

To sum up, it's my pleasure to recommend the current version of the manuscript for the publication after minor revisions.

Kind regards

Adam Emmer

---

## Author Response (AR2)

Response to Reviewer Comments, 3/21/2018

Minor specific comments and technical notes: *Responses in blue italics*

P4L29: technically, the mitigation measure applied at the dam of Lake Palcacocha is not a tunnel, but the combination of open cut and artificial dam

*The text in the manuscript has been changed to read: "A constant lake level of 4562 m (8 m of freeboard) is maintained by a structure in the smaller terminal moraine that consists of an open cut in a portion of the moraine filled with an artificial dam that was constructed in 1974 (Reynolds, 2003; Portocarrero, 2014)."*

P6L4: Mergili et al. (2017) is not in the list of references; r.avaflow has already been sucessfully employed to simulate GLOF process chain in the Cordillera Blanca (2012 event in Santa Cruz; see also http://onlinelibrary.wiley.com/doi/10.1002/esp.4318/full)

*Mergili et al. (2017) has been added to the reference list.*

P16L20: 'as the water surface level is lowered, the total volume stored in the lake increases' - please check

*The manuscript has been changed to read: "as the water surface level is lowered, the total volume stored in the lake **decreases**"*

P21L19-21: some passages are repeating, please check

*We checked for and corrected any repetition.*

P21L31: cannot be quantified using given approach

[revised manuscript text omitted]